# A proximity proteomics pipeline with improved reproducibility and throughput

Xiaofang Zhong [1,2,3,7], Qiongyu Li[1,2,3,7], Benjamin J Polacco[1,2,3], Trupti Patil[1,2,3], Aaron Marley[4], Helene Foussard[1,2,3], Prachi Khare[1,2,3], Rasika Vartak[1,2,3], Jiewei Xu[1,2,3], Jeffrey F DiBerto[5], Bryan L Roth [5], Manon Eckhardt [1,2,3], Mark von Zastrow [1,3,4], Nevan J Krogan [1,2,3] & Ruth Hüttenhain [1,2,3,6]✉

## Abstract

**Proximity labeling (PL) via biotinylation coupled with mass spectrometry (MS) captures spatial proteomes in cells. Large-scale processing requires a workflow minimizing hands-on time and enhancing quantitative reproducibility. We introduced a scalable PL pipeline integrating automated enrichment of biotinylated proteins in a 96-well plate format. Combining this with optimized quantitative MS based on data-independent acquisition (DIA), we increased sample throughput and improved protein identification and quantification reproducibility. We applied this pipeline to delineate subcellular proteomes across various compartments. Using the 5HT$_{2A}$ serotonin receptor as a model, we studied temporal changes of proximal interaction networks induced by receptor activation. In addition, we modified the pipeline for reduced sample input to accommodate CRISPR-based gene knockout, assessing dynamics of the 5HT$_{2A}$ network in response to perturbation of selected interactors. This PL approach is universally applicable to PL proteomics using biotinylation-based PL enzymes, enhancing throughput and reproducibility of standard protocols.**

**Keywords** Proximity Proteomics; APEX2-based Proximity Labeling; Protein–Protein Interaction; Subcellular Proteomics; G Protein-Coupled Receptor
**Subject Categories** Methods & Resources; Proteomics; Signal Transduction

## Introduction

The subcellular localization of proteins and their interactions with other proteins play a pivotal role in protein function. Both protein localization and protein–protein interactions (PPIs) are dynamic processes, with proteins navigating between subcellular neighborhoods and altering their interaction networks to carry out their functions in response to extracellular signals. Therefore, systematic characterization of protein localization and dynamic protein networks is essential for elucidating the molecular mechanisms underlying fundamental cellular processes.

Proximity labeling (PL) has emerged as a robust and versatile proteomics approach for globally mapping the spatial organization of proteins (Rhee et al, 2013; Go et al, 2021) and their molecular interactions, including protein–protein (Go et al, 2021; Lobingier et al, 2017; Paek et al, 2017), protein–RNA (Ramanathan et al, 2018; Qin et al, 2021b), and protein–DNA interactions (Gao et al, 2018; Myers et al, 2018). PL relies on genetically fusing an enzyme to a target protein or a protein localization domain that subsequently catalyzes the proximity-dependent biotinylation of biomolecules in live cells. Various PL enzymes have been developed representing two main enzyme classes: (1) peroxidases, including horseradish peroxidase (Li et al, 2014; Martell et al, 2016) and engineered ascorbate peroxidase 2 (APEX2) (Lam et al, 2015) and (2) biotin ligases including BioID (Roux et al, 2012), UltraID (Kubitz et al, 2022), TurboID and miniTurbo (Branon et al, 2018), and others (Samavarchi-Tehrani et al, 2020; Qin et al, 2021a).

The underlying principle of all these PL approaches involves expressing the PL enzyme fusion construct in cells. To mitigate artifacts from overexpression and variability in fusion construct expression levels, most studies aim for moderate and inducible expression in isogenic cell clones (Samavarchi-Tehrani et al, 2018) or fusing the PL enzyme to the endogenously expressed proteins (Vandemoortele et al, 2019; Gupta et al, 2018), requiring the characterization of multiple cellular clones for each PL construct. Once a suitable cellular clone is selected, the biotinylation reaction might be performed under various experimental conditions to label proteins proximal to the protein of interest. Subsequently, biotinylated proteins are enriched for each sample using streptavidin beads, followed by a series of washing steps to remove non-biotinylated proteins and proteolytic digestion to prepare samples for quantitative mass spectrometric (MS) analysis. However, this multi-step process is time-consuming and lacks scalability, thereby limiting sample throughput in current PL proteomics applications. Moreover, sample preparation across multiple batches may introduce variability, compromising reproducibility in quantifying

[1]Quantitative Biosciences Institute (QBI), University of California, San Francisco, San Francisco, CA 94158, USA. [2]J. David Gladstone Institutes, San Francisco, CA 94158, USA. [3]Department of Cellular and Molecular Pharmacology, University of California, San Francisco, San Francisco, CA 94158, USA. [4]Department of Psychiatry and Behavioral Sciences, University of California, San Francisco, San Francisco, CA 94158, USA. [5]Department of Pharmacology, School of Medicine, University of North Carolina at Chapel Hill, Chapel Hill, NC 27599, USA. [6]Department of Molecular and Cellular Physiology, Stanford University School of Medicine, Stanford, CA 94305, USA. [7]These authors contributed equally: Xiaofang Zhong, Qiongyu Li. ✉E-mail: ruthh@stanford.edu

biotinylated proteins across batches. Although PL represents a powerful technology to study dynamics in subcellular localizations as well as protein interactions across diverse conditions, throughput can be a limiting factor due to the multi-step nature of the standard protocol (Go et al, 2021; Antonicka et al, 2020; Göös et al, 2022; Piette et al, 2021).

To address these issues, we present a scalable and reproducible pipeline for PL-based proteomics (Fig. 1), incorporating automated enrichment of biotinylated proteins in a 96-well plate format and an optimized proteomics method for reproducible protein quantification. To facilitate PL studies with higher throughput, we developed an automated 96-well format platform for enriching biotinylated proteins using magnetic streptavidin beads on the KingFisher Flex platform, followed by enzymatic digestion and sample clean up in a 96-well format manually. While PL samples have commonly been analyzed using data-dependent acquisition (DDA)-based MS, the preference for high-abundance peptides in DDA can result in stochastic and irreproducible measurements (Aebersold and Mann, 2003; Michalski et al, 2011; Hu et al, 2016). In contrast, owing to the systematic and unbiased fragmentation of all precursor ions within a predefined mass range, data-independent acquisition (DIA) achieves higher quantitative consistency at comparable proteome coverage (Gillet et al, 2012; Collins et al, 2017; Ludwig et al, 2018). Therefore, we optimized a DIA-MS method for samples derived from PL experiments based on several parameters, including precursor scan range, normalized collision energy, maximum injection time, and the precursor isolation window overlap. The combination of automated enrichment for biotinylated proteins with the DIA-MS method not only increased throughput, but also provided higher reproducibility of protein identification and quantification compared to manual sample processing.

To evaluate our strategy, we selected two common PL applications: mapping subcellular proteomes and quantitatively analyzing protein interaction network dynamics. First, we used APEX2-based PL to identify compartment-specific proteins for several subcellular compartments including plasma membrane, endosome, Golgi apparatus, and lysosome. Our approach successfully identified compartment-specific proteins aligned with their expected locations based on a comparison with the Human Cell Map (Go et al, 2021). Next, we applied our PL strategy to map temporal protein interaction network changes of the $5HT_{2A}$ serotonin receptor, a G protein-coupled receptor (GPCR), following activation with its endogenous ligand serotonin (5-

hydroxytryptamine; 5HT). Our PL approach not only achieved high reproducibility in quantifying ligand-induced proximal proteome changes over a time course but also identified well-known regulators of GPCR function, such as G protein-coupled receptor kinase 2 (GRK2), beta-arrestin 2, as well as subunits of protein kinase C and D (PKC and PKD, respectively). Overall, the activity-dependent protein interaction changes for the $5HT_{2A}$ receptor could be clustered into rapid events with transient changes and later events with sustained differences in biotin labeling.

Finally, we assessed the applicability of our PL pipeline for capturing proximal protein network dynamics across multiple experimental conditions. As high-throughput PL studies are not only limited by the enrichment of biotinylated proteins, but also by the scale of the cell culture required to generate sufficient sample input amounts, we initially adapted our PL pipeline to accommodate low sample input amounts. We demonstrated that PL experiments could be performed in a 6-well plate format without compromising performance substantially compared to the high-input PL. To test the potential of our method for investigating interaction dynamics across multiple conditions, we then utilized CRISPR-based gene knockout to perturb selected components of the $5HT_{2A}$ interaction network and to monitor resulting activation-dependent PL dynamics with the low-input PL. Our data revealed that the knockout of both, Annexin A2 (ANXA2) and S100-A10 (S100A10), two components of the $5HT_{2A}$ network, resulted in reduced biotinylation of ANXA2, S100A10, and AHNAK, which is consistent with previous findings demonstrating that these proteins physically interact (Benaud et al, 2004; Chen et al, 2022).

In summary, we described a versatile PL pipeline applicable across a wide range of PL applications, various experimental scales, and multiple experimental conditions. Although we focus on APEX2-based PL in our study, the automated enrichment of biotinylated proteins can be widely utilized to accelerate essentially any PL strategy reliant on biotinylation.

## Results

### Generating monoclonal cell lines for proximity labeling

To set up a PL proteomics experiment, the PL enzyme is typically genetically fused to a protein localization domain (PLD) or a

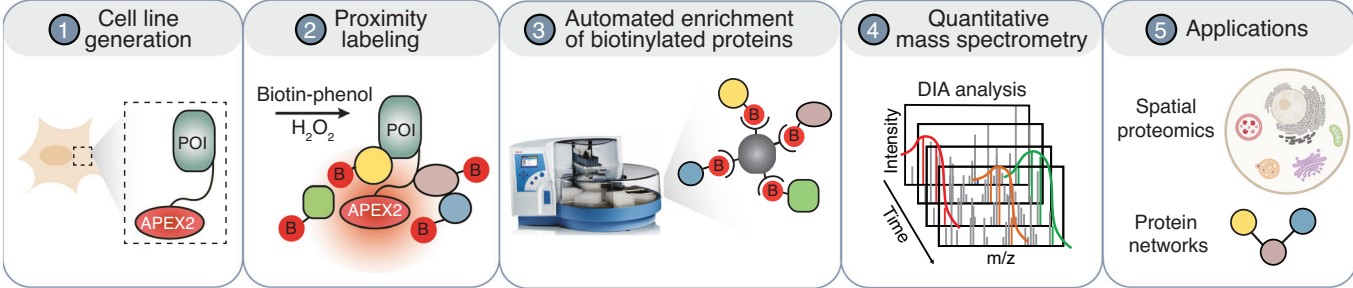

**Figure 1.  A schematic overview of universal workflow for proximity proteomics.**

This workflow includes (1) PL construct design as well as generation and characterization of monoclonal cell lines, (2) proximity labeling, (3) automated enrichment of biotinylated proteins, (4) quantitative mass spectrometry data acquisition, and (5) the application of this workflow to map subcellular proteomics as well as protein–protein interactions and their dynamics.

protein of interest (POI) and expressed in cells. Here, we present considerations for designing PL constructs and outline the process of generating and characterizing monoclonal cell lines tailored for PL proteomics. To generate cell lines with stable and inducible expression of the PL constructs, we introduced APEX2 constructs into a lentiviral backbone under a Doxycycline-inducible promoter containing Puromycin resistance for antibiotic selection (Fig. 2A). In addition, the constructs were appended with a FLAG epitope and/or GFP for detection via fluorescence-activated cell sorting (FACS), imaging, and western blot analysis. For spatial proteomics, a PLD was selected to direct the PL construct to the subcellular compartment of interest. For protein interaction mapping, the APEX2 enzyme was fused to the POI and separated by a flexible linker to minimize interference of the PL enzyme with the protein's function (Fig. 2A). After generating polyclonal cells by lentiviral transduction and puromycin selection, the expression of PL constructs was validated using FACS and immunofluorescence microscopy. To achieve consistent and homogeneous expression of the APEX2-tagged constructs and to mitigate shifts in the cellular populations between biological replicates, monoclonal cell lines were generated by single-cell sorting. To identify optimal clones after sorting, multiple clones were initially screened using flow cytometry to assess uniformity of the fusion construct expression. Promising clones were further evaluated by confocal immunofluorescence imaging to validate subcellular localization of the fusion construct, followed by western blot analysis to evaluate their biotinylation capacity (Fig. 2A).

As an illustration, we detailed the generation and characterization of monoclonal cell lines in HEK293T cells for a plasma membrane (PM) localized APEX2 construct, for which the enzyme was directed to the PM using a Lyn11 motif appended with GFP for visualization (Appendix Fig. S1, Dataset EV1). Following lentiviral transduction and puromycin selection, the polyclonal cell line exhibited heterogeneous expression of the PL construct, with varied GFP expression patterns observed, such as GFP expression in cytoplasm and nucleus in addition to the PM (Fig. 2B). Following single-cell sorting, 24 monoclonal PM-APEX2 cell lines were subjected to flow cytometry analysis to assess expression levels and consistency of the APEX2-tagged PM construct based on GFP fluorescence. Monoclonal cell lines containing over 85% GFP-positive cells were prioritized for further characterization by confocal immunofluorescence microscopy (Fig. 2C). Confocal imaging revealed distinct localization patterns for different PM-APEX2 clones, with clones 7, 8, and 11 exhibiting concentration at the PM, while clone 1 displayed mainly diffuse cytoplasmic distribution (Fig. 2D). Finally, western blot analysis was performed to evaluate the biotinylation efficiency of the monoclonal PM-APEX2 cell lines. Biotinylation was initiated by addition of biotin-phenol and $H_2O_2$, with the sole biotin-phenol treatment as the negative control. All PM-APEX2 clones demonstrated varying levels of biotin labeling (Fig. 2E). Considering unspecific labeling associated with higher biotinylation efficiency, clones 8 and 11, exhibiting moderate biotinylation efficiency, were selected for PL studies.

Following this workflow, monoclonal cell lines could be generated and characterized within a timeframe of three weeks. Although APEX2-based PL was employed in our study, these experimental procedures are adaptable to any other PL enzyme.

## A scalable biotin enrichment method coupled with quantitative MS

The enrichment of biotinylated proteins represents a key step in PL workflows. However, manual enrichment using streptavidin beads, followed by multiple washing steps to remove non-biotinylated proteins, is not only time-consuming but can also introduce variability between samples, thereby compromising quantitative reproducibility. To address these challenges and facilitate large-scale PL studies while maintaining high reproducibility and quantitative accuracy, we devised a PL strategy combining automated enrichment of biotinylated proteins in 96-well plate format with MS analysis using data-independent acquisition (DIA) (Fig. 3A). We developed an automated enrichment protocol using magnetic streptavidin beads to bind biotinylated proteins, adapted for use on the KingFisher Flex system (Fig. 3B). The protocol, outlined in Dataset EV2, requires less than 5 min to load the corresponding plates into the system.

To maximize the enrichment efficiency, we evaluated the impact of varying protein input amounts. Using the PM-APEX2 cell line, we compared protein input amounts ranging from 0.5 to 2 mg for the enrichment. Our results indicated minor differences comparing 0.5, 1, and 2 mg in the number of peptides and proteins identified, with 1 mg yielding slightly better performance (Appendix Fig. S2A), suggesting the protocol's applicability across a range of protein input amounts. In addition, we tested different volumes of streptavidin beads (60, 80, and 100 μl) while maintaining a constant protein input amount of 1 mg. To evaluate nonspecific protein binding to beads, we performed the APEX experiment for the PM-APEX2 cell line in the presence or absence of $H_2O_2$. Sixty μl of streptavidin bead slurry resulted in significantly lower peptide and protein identifications compared to 80 and 100 μl beads (Fig. 3C). Notably, similar numbers of proteins were identified for 80 μl and 100 μl bead volumes with the presence or absence of $H_2O_2$ (Fig. 3C; Appendix Fig. S2B), with the 80 μl bead volume condition sacrificing peptide and protein IDs only minimally while achieving slightly lower nonspecific protein binding for the smaller bead volume. These results suggest that there is a wider optimum for the bead volume to be used for PL experiments. However, with a higher volume of beads, the intensities of streptavidin peptides released during proteolytic digestion also increased (Appendix Fig. S2C, Dataset EV3), which might interfere with identification of biotinylated proteins. Based on these observations, we opted for 80 μL of streptavidin beads in subsequent experiments, considering both cost-effectiveness and performance. It is important to note that the optimal starting protein lysate amount and bead volume may depend on factors such as PL construct expression levels, cell types, and bead types.

We further aimed to enhance the throughput by testing shorter binding times for biotinylated proteins to streptavidin beads. While 1 and 2 h binding time seemed insufficient, minor differences were observed between 4 and 18 h in the number of peptides and proteins identified, indicating the feasibility of completing the enrichment of biotinylated proteins within a single day (Fig. 3D).

To maximize protein identifications and quantitative accuracy of the PL strategy, we fine-tuned several parameters for the MS analysis of PL samples using DIA, including precursor scan range, normalized collision energy, maximum injection time, and the

overlap width of the precursor isolation windows. Optimized parameters for the DIA method consisted of one full MS scan over a range of 350–1050 $m/z$ (Appendix Fig. S2D), MS[2] scans with a normalized HCD (higher-energy collisional dissociation) collision

energy of 30% (Appendix Fig. S2E), a maximum injection time of 32 ms (Appendix Fig. S2F), and isolation windows of 20 $m/z$ over 350–1050 $m/z$ range with 2 $m/z$ overlap between windows (Appendix Fig. S2G).

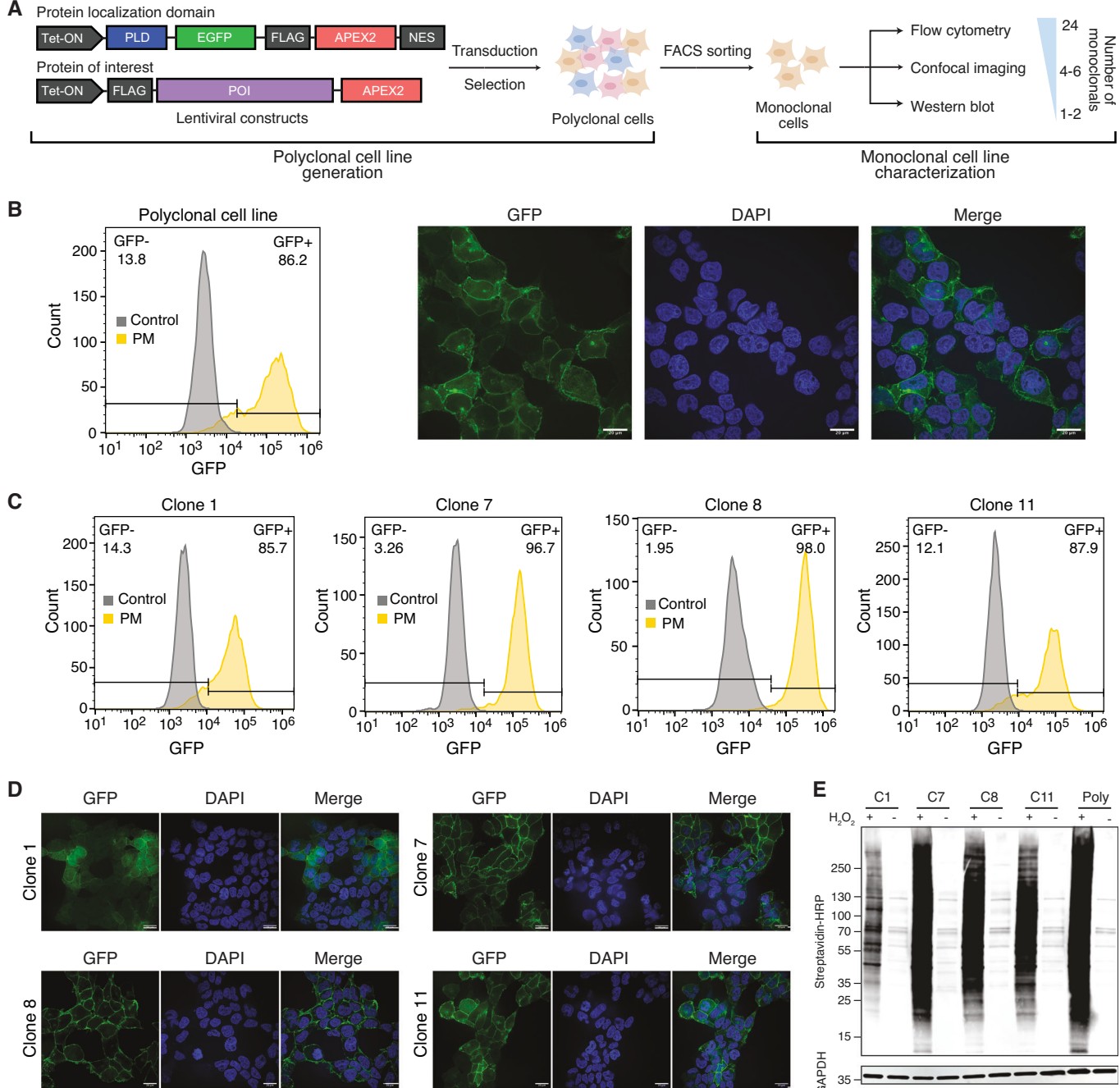

**Figure 2. Step-by-step guide for generating monoclonal cell lines for proximity labeling.**

(**A**) Schematic overview of PL construct design and generation and stepwise characterization of monoclonal cell lines for PL (PLD refers to protein localization domain and POI to protein of interest). (**B**) Characterization of polyclonal proximity labeling cell lines by flow cytometry (left) and confocal imaging (right). The APEX2 enzyme was targeted to the plasma membrane using a Lyn11 protein localization domain (PM-APEX2). GFP was used to evaluate expression and localization of the APEX2 construct. The non-Doxycycline induced cell line was used as control for flow cytometry. Scale bar represents 20 μm. (**C**) Flow cytometry analysis of four monoclonal PM-APEX2 cell lines. (**D**) Confocal imaging of four monoclonal PM-APEX2 cell lines. Cells were stained with DAPI and Hoechst 33342 for the nucleus and the APEX2 construct location was indicated by GFP. Scale bar represents 20 μm. (**E**) Western blot analysis of whole cell lysate derived from four monoclonal PM-APEX2 cell lines to evaluate the biotin labeling efficiency of APEX2 using streptavidin antibody. GAPDH is used as loading control.

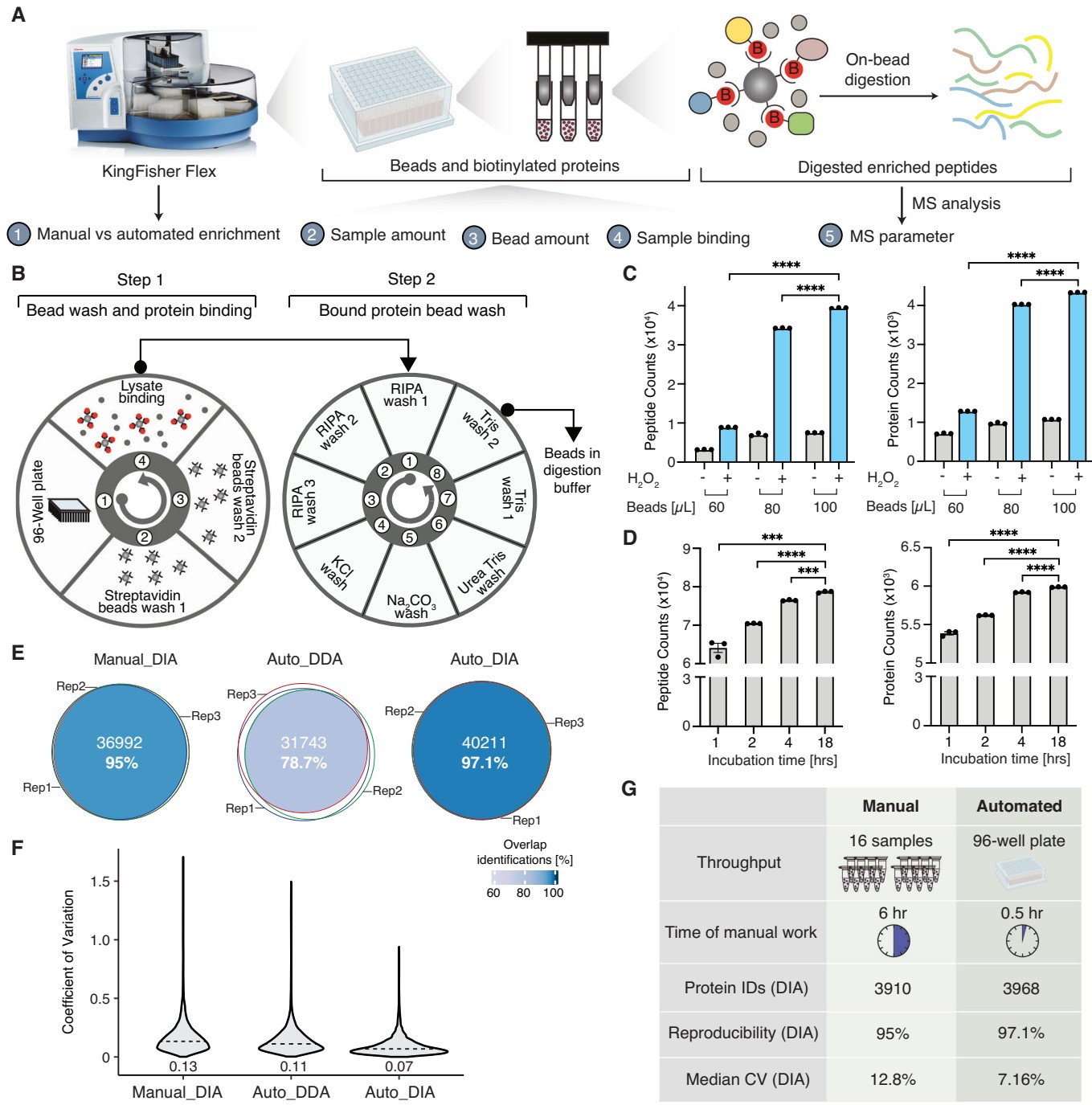

**Figure 3. A scalable biotin enrichment method coupled with quantitative MS.**

(A) Schematic overview of the development and optimization of the automated enrichment method for biotinylated proteins and quantitative mass spectrometry method. (B) Overview of the automated biotin enrichment protocol established on the KingFisher Flex. (C) Peptides and proteins identified for varying amounts of streptavidin beads (60, 80, and 100 µl) in the presence and absence of $H_2O_2$ treatment to assess nonspecific protein binding to beads depending on the bead amount, $n = 3$ biological replicates, statistical significance was determined using unpaired t-test in Prism (GraphPad). Results are presented as mean ± standard error of the mean (SEM) (Peptides: 60 µL vs. 100 µL, $p < 0.0001$; 80 µL vs. 100 µL, $P < 0.0001$. Proteins: 60 µL vs. 100 µL, $p < 0.0001$; 80 µL vs. 100 µL, $P < 0.0001$). (D) Peptides and proteins identified for varying binding times (1, 2, 4, and 18 h) of biotinylated proteins to streptavidin beads, $n = 3$ biological replicates, statistical significance was determined using unpaired t-test in Prism (GraphPad). Results are presented as mean ± standard error of the mean (SEM) (Peptides: 1 h vs 18 h, $p < 0.001$; 2 vs 18 h, $p < 0.0001$; 4 vs 18 h, $p < 0.001$. Proteins: 1 h vs 18 h, $p < 0.0001$; 2 vs 18 h, $p < 0.0001$; 4 vs 18 h, $p < 0.0001$). (E) Performance comparison of manual enrichment protocol combined with data-independent acquisition MS (Manual_DIA), automated enrichment combined with data-dependent acquisition MS (Auto_DDA), and automated enrichment protocol with data-independent analysis (Auto_DIA). Performance was evaluated comparing the precursor ions being identified from three biological replicates ($n = 3$). (F) Distribution of the coefficient of variation of protein intensities from three replicates comparing Manual_DIA, Auto_DDA, and Auto_DIA approaches. Numbers depicted on the graph represent median CVs. (G) Summary comparison of the manual and automated enrichment method for biotinylated proteins.

Finally, to assess the protein identification and quantification capabilities of the optimized DIA method coupled with the automated enrichment method (Auto_DIA), we compared it with manual biotinylation enrichment coupled with DIA analysis (Manual_DIA) as well as a data-dependent analysis (DDA) method with automated enrichment (Auto_DDA). Auto_DIA outperformed Manual_DIA and DDA analysis (Auto_DDA) based on the following results: (1) The number of precursor ions identified across all the three replicates were 36,992 (95%) for Manual_DIA, 31,743 (78.7%) for Auto_DDA, and increased to 40,211 (97.1%) for Auto_DIA (Fig. 3E). Similar trends were also observed for the peptides and proteins identified across three independent replicates (Appendix Fig. S3A–C, Appendix Table S1); (2) Auto_DIA achieved a narrower distribution of coefficients of variation (CV) for protein intensities across three independent replicates with median CV of 7%, while Manual_DIA and Auto_DDA resulted in median CVs of 13% and 11%, respectively (Fig. 3F); (3) The correlation coefficients comparing peptide intensities of the three replicates ranged from 0.91 to 0.948 for Manual_DIA, 0.917-0.95 for Auto_DDA, but improved to 0.941–0.967 for Auto_DIA (Appendix Fig. S3D).

Collectively, we established a PL strategy integrating an automated protocol for biotinylated protein enrichment coupled with optimized DIA MS analysis, enabling high-throughput sample processing in 96-well plate format, significantly reduced manual processing time, and enhanced reproducibility in terms of protein identification and quantification across replicates (Fig. 3G).

## Mapping subcellular proteomes using the automated proximity proteomics strategy

PL is a premier method for spatial proteomics and delineating the proteome composition of subcellular structures. To evaluate the performance of our optimized proximity proteomics strategy in subcellular proteome mapping, we generated and characterized monoclonal HEK293T cell lines expressing APEX2 proximity labeling constructs targeted to the endosome, Golgi apparatus, lysosome, and plasma membrane using the step-by-step approach outlined above. The protein localization domains included: Lyn11 for the plasma membrane, 2xFYVE for the endosome, LAMTOR1 targeting sequence and full-length LAMP1 for the late endosome/ lysosome, β-1,4-galactosyltransferase (GalT) for the Golgi apparatus, and nuclear export signal (NES) as control (Appendix Fig. S1, Dataset EV1). The final selected clones exhibited homogenous expression of the APEX2 construct with over 92% GFP-positive cells, except for the NES-APEX2 construct (Appendix Fig. S4A). Furthermore, the APEX2 constructs localized to the corresponding subcellular compartments of interest were confirmed by co-localization analysis with markers, including Vimentin for cytosol, E-cadherin for plasma membrane, Rab5 for early endosome, Rab9 for late endosome/lysosome, and Golgin-97 for Golgi apparatus (Fig. 4A). Lastly, the final selected clones demonstrated median biotinylation intensities and minimal background biotinylation among other monoclonal cell lines (Appendix Fig. S4B), which varied among different clones, likely due to the distinct expression levels of APEX2-tagged localization constructs (Appendix Fig. S4C). Following biotinylation, proteins underwent automated enrichment, on-bead proteolytic digestion, and quantitative DIA-MS. All cell lines showed successful biotinylation and efficient enrichment, as assessed by western blot analysis (Appendix Fig. S5).

Principal component analysis (PCA) comparing the biotinylated proteins for each localization construct showed separation of each subcellular structure, but clustering of their biological replicates (Fig. 4B). To determine compartment-specific proteins, we performed a two-step filtering process. First, protein abundances for each location construct were compared to the control (NES-APEX2) using MSstats (Choi et al, 2014) (Fig. 4C; Appendix Fig. S6). All proteins with a $\log_2$ fold change >1 and adjusted $p$-value < 0.05 were considered candidate compartment-specific proteins (Dataset EV4). Second, significant proteins were required to be exclusively present in one comparison, except for the proximal proteomes of the endosome (2xFYVE) and lysosome/late endosome (LAMP1, and LAMTOR1) due to an expected significant overlap in their subcellular proteomes. Based on these criteria, 337, 55, and 228 location-specific proteins were identified for the endosome and lysosome/late endosome (Endo-Lyso), Golgi apparatus, and plasma membrane, respectively (Fig. 4D). Based on the median normalized $\log_2$FC (fold change) across all samples, the location-specific proteins were divided into three groups (Fig. 4E). Gene ontology enrichment analysis demonstrated that, as expected, proteins specific for 2xFYVE, LAMP1, and/or LAMTOR1 in Group 1 were enriched for the early endosome, endosome, and lysosome; Group 2, comprising proteins specific for GalT, showed enrichment for Golgi apparatus subcompartment and Golgi membrane, while proteins in group 3, specific for Lyn11, were enriched in actin filament-based process, anchoring junction, and plasma membrane region.

Finally, we compared the location-specific proteins derived from our PL analysis to the proteins defined for the respective compartments in the Human Cell Map (Go et al, 2021), a proximity map of the HEK293 proteome generated using BioID-based PL. As anticipated, proteins specific for 2xFYVE, LAMP1, and LAMTOR1 were enriched for the endosome and lysosome proteome determined by the Human Cell Map, while proteins specific for GalT were enriched in the Golgi apparatus, and Lyn11-specific proteins were enriched in the plasma membrane and cell junction (Fig. 4F). Notably, our dataset also uncovered location-specific proteins not identified in the Human Cell Atlas, such as Golgin-45 (Short et al, 2001), sphingomyelin syntheses 2 (Villani et al, 2008), and palmitoyltransferase ZDHHC14 (Adachi et al, 2016) for Golgi apparatus; cell surface hyaluronidase (Yamamoto et al, 2017), pleckstrin homology domain-containing family O member 1 (Olsten et al, 2004), and guanine nucleotide-binding protein G(i) subunit alpha-1 (Cho and Kehrl, 2007) for plasma membrane; and proteins involved in endosomal trafficking, like Ras-related protein Rab-4A (RAB4A) (Nag et al, 2018), charged multivesicular body protein 2a (CHMP2A) (Yang et al, 2022), sorting nexin-11 (SNX11) (Xu et al, 2013), sorting nexin-16 (SNX16) (Hanson and Hong, 2003), and vacuolar protein sorting-associated protein 41 homolog (VPS41) for endosome/lysosome (Bowers and Stevens, 2005). Taken together, employing our automated PL strategy enabled the identification of location-specific proteins in various cellular compartments, underscoring its applicability to spatial proteomics.

## Mapping the activity-dependent proximal interaction network of the 5HT$_{2A}$ receptor

Since PL is also applicable to mapping proximal protein networks of a protein of interest (POI), we sought to evaluate our PL pipeline

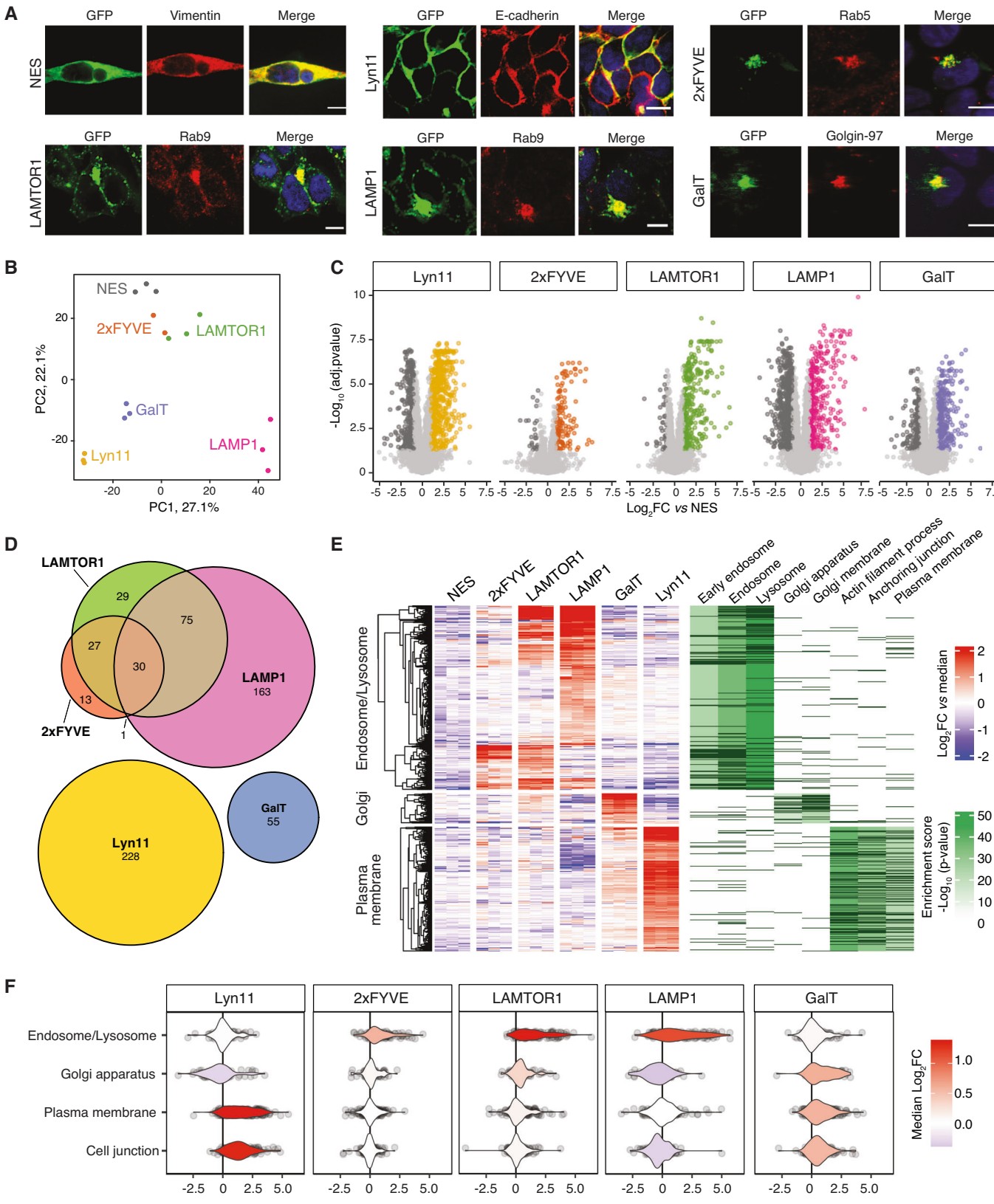

◀ **Figure 4. Mapping subcellular proteomes using the automated proximity proteomics strategy.**

(A) Confocal fluorescence imaging of monoclonal cell lines of all spatial APEX2 constructs to co-localize with a subcellular location marker. APEX2-fusion constructs were stained using a GFP antibody. Location markers of vimentin, E-cadherin, Rab5, Rab9, and Golgin-97 were used for NES (cytosol), Lyn11 (plasma membrane), 2xFYVE (endosome), LAMTOR1 and LAMP1 (late endosome/lysosome), and GalT (Golgi apparatus), respectively (Scale bars, 10 μm). (B) Principal-component analysis (PCA) comparing proximal proteomes of each replicate of the spatial APEX2 constructs, $n = 3$ biological replicates (except for 2xFYVE, $n = 2$). (C) Volcano plots comparing proximal proteome of each location specific construct to the control (NES). Proteins with a $\log_2$ fold change >1 and adjusted $p$-value < 0.05 were considered significant and colored. Data were collected from independent biological replicates $n = 3$ (except for 2xFYVE, $n = 2$), statistical analysis was performed using the statistical framework implemented in MSstats (Choi et al, 2014), $p$-values were adjusted using Benjamini–Hochberg. (Benjamini and Hochberg, 1995). (D) Venn diagram comparing the location-specific proteins. (E) Heatmap of location-specific proteins was highlighted as three groups based on the median normalized $\log_2$ fold change. Gene ontology (GO) enrichment analysis was performed for each cluster and the matched genes for each GO term are indicated in dark purple, $n = 3$ biological replicates (except for 2xFYVE, $n = 2$). (F) Comparison of the location proteins candidates identified in this study with the protein localizations defined in Human Cell Map (Go et al, 2021). For the analysis we used the following number of proteins representing the different subcellular compartments: 700 for endosome/lysosome, 543 for Golgi apparatus, 1352 for plasma membrane, and 953 for cell junction. $n = 3$ biological replicates (except for 2xFYVE, $n = 2$).

for studying dynamics of proximal protein networks. As a proof of concept, we selected the 5HT$_{2A}$ serotonin receptor, a member of the G protein-coupled receptor (GPCR) family, to map its proximal interaction network changes upon activation by its endogenous ligand, serotonin (5-hydroxytryptamine; 5HT). Despite the therapeutic relevance of 5HT$_{2A}$ (Kwan et al, 2022; McClure-Begley and Roth, 2022), beyond a subset of known signal transducers and regulatory molecules, little is known about the interaction network in response to 5HT$_{2A}$ activation. APEX-based PL allows the systematic mapping of its proximal interaction network upon receptor activation, which could unveil novel proteins regulating the receptor's activity that might in the future provide additional targets to finetune receptor activity.

To generate the 5HT$_{2A}$ PL construct, we placed the APEX2 enzyme within the C-terminal tail of the receptor to avoid disturbing the C-terminal PSD-95/Discs-large/ZO-1 (PDZ)-binding domain, which is essential for receptor-mediated signaling and trafficking (Xia et al, 2003) (Appendix Fig. S7A). Following monoclonal HEK293T cell line generation, we validated that the 5HT$_{2A}$-APEX2 construct localized to the plasma membrane (Appendix Fig. S7B) and retained signaling activity determined based on Gq heterotrimer dissociation (Appendix Fig. S7C). For PL, the receptor was activated with 5HT over a 30-minute time course in the presence of biotin phenol and H$_2$O$_2$, followed by automated enrichment of biotinylated proteins and DIA-MS (Fig. 5A). Our optimized DIA-MS method quantified 6429 proteins across the dataset. To identify proteins with agonist-dependent labeling changes, we performed an analysis of variance (ANOVA) by fitting a polynomial curve for each protein across the time course (Fig. 5A), identifying 744 proteins with significant differences following 5HT treatment (adjusted $p$-value < 0.05) (Fig. 5B, Dataset EV5). An overrepresentation analysis of location-specific proteins for plasma membrane, endosome, late endosome/lysosome (Fig. 4D) suggested a decrease in biotinylation of plasma membrane-specific proteins and an increase of endosome and lysosome-specific proteins across the time-course, indicating ligand-induced receptor trafficking (Fig. 5B). To distinguish between (1) location-specific proteins residing in the receptor's local environment but not participating in its function and (2) protein networks potentially regulating receptor signaling and trafficking, we examined the fold change (FC) distribution of the location-specific proteins and further filtered the significant proteins with FC > 1.5 (Fig. 5C). Based on the temporal profiles, the remaining 142 proteins formed two distinct clusters: a cluster

representing proteins with rapid kinetics and transient changes in biotin labeling at 1 min after agonist treatment and the second cluster with later, sustained differences in labeling across 30 min of 5HT treatment (Fig. 5D).

The transient cluster was enriched for proteins functioning in GPCR signaling and Rho GTPases signaling (Fig. 5E), including well-characterized interactors of 5HT$_{2A}$ such as G protein-coupled receptor kinase 2 (GRK2), which phosphorylates 5HT$_{2A}$, and beta-arrestin 2 (ARRB2), which binds to the phosphorylated receptor to meditate receptor desensitization and internalization (Allen et al, 2008). We also observed increased transient labeling of downstream 5HT$_{2A}$ signaling targets such as protein kinase C (PKCD, PKCA, and PKCQ; isoforms of protein kinase C), protein kinase D (PRKD2), and diacylglycerol kinase zeta (DGKZ), which have long been implicated in 5HT$_{2A}$ signaling (Roth and Chuang, 1987; Allen et al, 2008). Moreover, 5HT$_{2A}$ activation triggered the signaling by Rho GTPases, which play a key role in the regulation of actin dynamics through activation of WASPs (Wiskott-Aldrich syndrome proteins) and WAVEs (WASP family verprolin-homologous) (Sit and Manser, 2011). Our dataset recovered four proteins of WASP/WAVE complex with 5HT-dependent increase in PL. While the transient cluster contained mainly proteins involved in receptor signaling, the sustained cluster showed enrichment of proteins involved in membrane trafficking (Appendix Fig. S7D). Specifically, we identified regulators of clathrin-mediated endocytosis including BMP-2 inducible kinase (BMP2K), the mannose 6-phosphate receptor (M6PR), as well as the clathrin adapter proteins Epsin-1 (EPN1) and Epsin-2 (EPN2) in our dataset (McMahon and Boucrot, 2011; Wang et al, 2016; Mettlen et al, 2018). Previous studies have demonstrated the involvement of clathrin components in 5HT$_{2A}$ receptor endocytosis (Bhatnagar et al, 2001). The results from the 5HT$_{2A}$ receptor study suggest that our PL strategy can capture transient and sustained dynamics of proximal protein networks. We envision that automated enrichment in 96-well plate format combined with DIA-based MS accelerates PL studies across many conditions in parallel and, given its quantitative reproducibility, allows for sensitive detection of PL changes.

## Exploring proximal interaction network perturbations induced by CRISPR-based gene knockout using low-input PL pipeline

Next, we sought to assess whether our PL pipeline is amenable to monitoring interaction network changes across multiple conditions. As a proof of concept, we wanted to examine how CRISPR-

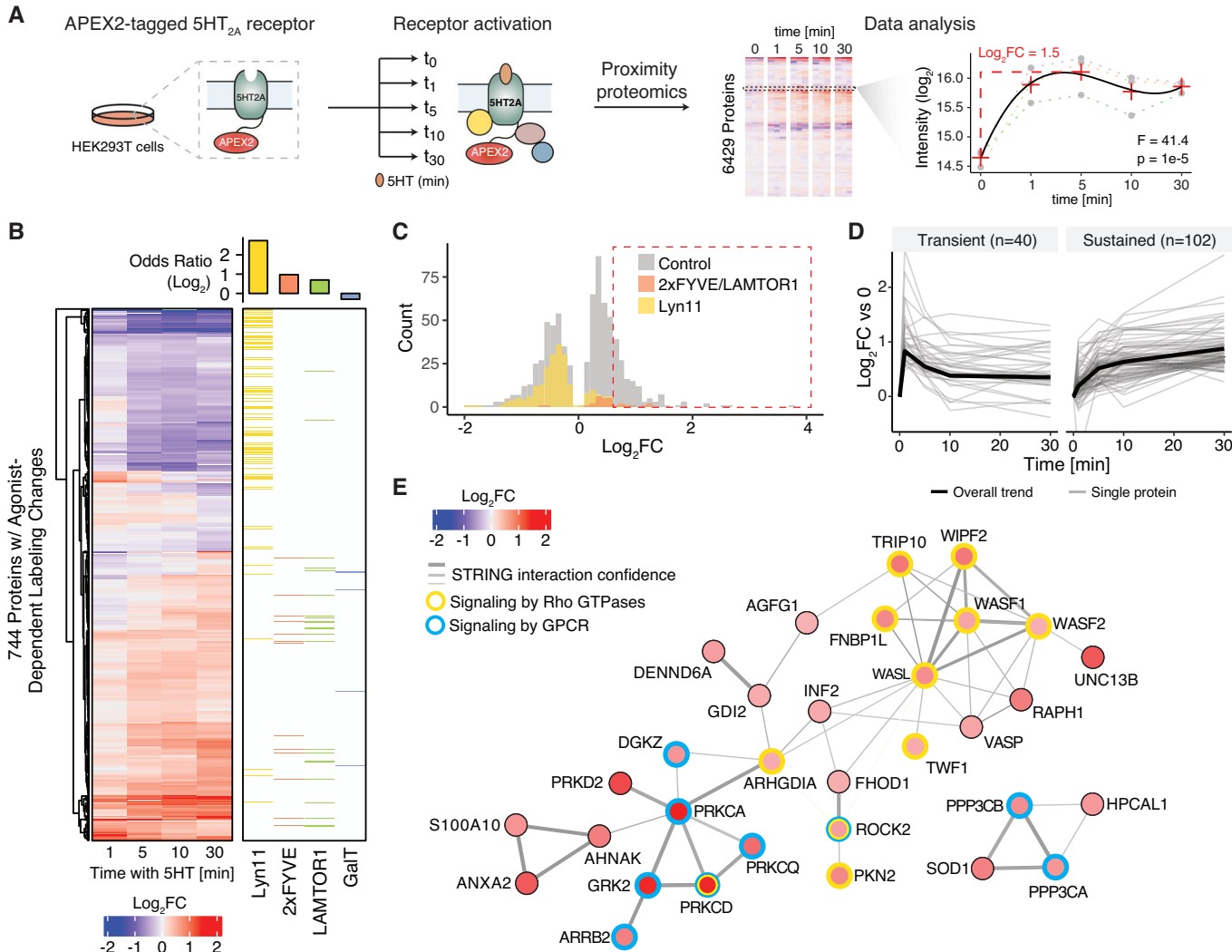

**Figure 5. Mapping the activity-dependent proximal interaction network of the 5HT₂ₐ receptor.**

(A) Experimental design to map proximal interaction networks of 5HT₂ₐ. APEX2-fused 5HT₂ₐ receptor was stably expressed in HEK293T cells, activated with 5HT and at indicated time points after agonist treatment proteins proximal of the receptor were biotinylated by addition of $H_2O_2$. Following automated enrichment of biotinylated proteins, the samples were analyzed using DIA and statistical analysis was performed by fitting a curve through the time points and calculating $p$ values using F-test statistics. (B) Heatmap depicting proteins that significantly change in biotinylation after agonist addition. Overrepresentation of location-specific proteins for plasma membrane (Lyn 11), endosome (2xFYVE), late endosome/lysosome (LAMTOR1), and Golgi (GalT) was shown in the right heartmap. Data were collected from three independent biological replicates, $n = 3$, statistical analysis was performed using ANOVA. (C) $Log_2$ fold change distribution of proteins in (B) (Control: Gray). Endosomal/lysosomal proteins (2xFYVE/LAMTOR1) and plasma membrane proteins (Lyn11) were highlighted in orange and yellow, respectively. The time point of each protein used for analysis is corresponding to the maximum $log_2$ fold change. The intensity at each timepoint was included in Dataset EV5. (D) Line charts of proteins filtered out from (C) with a fold change cutoff of 1.5 clustered into proteins with transient and sustained responses to 5HT treatment. Forty and 102 proteins are grouped into transient and sustained clusters, respectively. (E) Protein interaction network connecting proteins with transient agonist-dependent changes in the proximity of 5HT₂ₐ. Proteins shown as nodes were colored according to their $log_2$ fold change. The edges connecting the proteins were derived from STRING (Szklarczyk et al, 2011) and the edge width was scaled according to the interaction confidence. Proteins corresponding to Reactome pathways (Jassal et al, 2020) that were enriched within the transient cluster were indicated with different colored node borders.

based knockout (KO) of individual network components affects the overall activity-dependent 5HT₂ₐ interaction network (Fig. 6A). High-throughput PL studies across various conditions are not only limited by the enrichment of biotinylated proteins but also by the scale of the cell culture required to generate sufficient sample input amounts. Given the stable performance of the automated enrichment of biotinylated proteins across a range of protein input amounts and its high reproducibility (Fig. 3, Appendix Fig. S2), we

reasoned that our PL strategy (high-input PL) could be applicable to lower input amounts (low-input PL). To modify the high-input PL pipeline for lower sample inputs, we adjusted several parameters, including reducing cell culture scale from a 10-cm dish to a 6-well plate format, decreasing bead and reagent volumes throughout the enrichment and protein digestion, as well as increasing sample injection volume for MS analysis. All parameters for the low-input PL pipeline are listed in Appendix Table S2. Next,

we compared the performance of the high- and low-input PL pipeline based on reproducibility in protein identification and quantification following PL of the activated 5HT$_{2A}$-APEX. While the low-input PL protocol identified overall less peptide features, the protein identifications for the low-input protocol decreased by only 8.6% compared to the high-input (Fig. 6B). Moreover, the quantitative reproducibility was only marginally affected by the low-input PL protocol increasing the average coefficient of variation from 7.6% to 10.5% (Fig. 6C). These results suggest that it is indeed feasible to utilize the low-input protocol to perform PL studies without compromising performance substantially.

To examine how CRISPR-based KO of individual network components affects activation-dependent changes in the 5HT$_{2A}$ interaction network, we selected two proximal 5HT$_{2A}$ interactors from the transient cluster (Fig. 5D), ANXA2 (Annexin A2) and S100A10 (also known as p11). S100A10 and ANXA2 form a heterotetramer (Réty et al, 1999), which has been shown to bind to membrane proteins, such as the metabotropic glutamate receptor 5 (mGluR5) (Lee et al, 2015), L-type voltage-gated calcium channel (VGCC) (Jin et al, 2020), and a subset of the 5HT receptor family (Svenningsson et al, 2006; Warner-Schmidt et al, 2009) assisting their trafficking to the plasma membrane (Chen et al, 2022). To deplete endogenous ANXA2 and S100A10 by CRISPR-Cas9 KO, two guide RNAs (gRNA) were designed per gene, complexed with Cas9, and electroporated into 5HT$_{2A}$-expressing HEK293T cells (Fig. 6A). Two non-targeting control (NTC) gRNAs were included as negative controls. We demonstrated high KO efficiency in the polyclonal KO cell lines for both, S100A10 and ANXA2, as evaluated by inference of CRISPR edits or Western Blot analysis, respectively (Fig. 6D,E). Using the low-input PL protocol we performed APEX-based PL of the 5HT-activated 5HT$_{2A}$ receptor across multiple time points in different KO cell lines.

The majority of proteins in the transient (38/40) (Fig. 6F) and the sustained (89/102) (Appendix Fig. S8) clusters of the 5HT-evoked 5HT$_{2A}$ proximal proteome discovered with the high-input PL protocol were also captured by the low-input protocol. Notably, comparable kinetics were observed across the timepoints of 5HT treatment when comparing the high- and low-input PL protocols. Moreover, the data suggested a high level of consistency and reproducibility across knockout (KO) conditions (Fig. 6F, Appendix Fig. S8). In ANXA2-KO cells, there was a significant reduction in S100A10 protein levels compared to NTC cells, both with and without 5HT$_{2A}$ receptor activation. However, the converse knock-out of S100A10 did not significantly affect ANXA2 protein levels (Fig. 6G). In addition, in both S100A10-KO and ANXA2-KO cells, the protein AHNAK exhibited reduced proximal labeling across all conditions (Fig. 6G). Neither ANXA2 nor S100A10 KO resulted in significant changes in sustained, activity-dependent proximal interactions (Appendix Fig. S8). Finally, to investigate the effect of ANXA2 and S100A10 KO on cell surface expression of the 5HT$_{2A}$ receptor, we quantified the receptor pool at the plasma membrane using flow cytometry in KO and non-targeting control cells. The results indicated a small but significant decrease in the expression of the 5HT$_{2A}$ receptor at the plasma membrane in both ANXA2 and S100A10 KO cells (Appendix Fig. S9). This validates the potential of our method for identifying biologically relevant interactors and, specifically, suggests a role of the ANXA2-S100A10 complex in determining 5HT$_{2A}$ receptor trafficking or stability, but further functional validation is necessary to test this hypothesis and explore the functional significance of the novel interactions detected.

Together, our study demonstrates the efficacy of the automated, low-input PL protocol in delivering reproducible results at a smaller scale and in monitoring interaction network perturbations across multiple conditions with CRISPR KO of proximal interactors.

## Discussion

In this study, we present a scalable and reproducible approach for PL-based proteomics aimed at accelerating large-scale PL investigations (Fig. 1). Initially, we outlined a detailed protocol for the generation and characterization of monoclonal cell lines for PL (Fig. 2), a crucial step for ensuring consistent PL levels across diverse experimental conditions. Subsequently, we implemented an automated 96-well format method on the KingFisher Flex instrument for enriching biotinylated proteins, thus minimizing variability inherent in manual sample processing (Fig. 3). Compared to the labor-intensive 3-h manual enrichment process capable of handling up to around 16 samples, our automated approach enables processing of 96 samples in parallel with only 30 min of manual intervention. To optimize the automated protocol, we fine-tuned various parameters, including the amounts of protein lysate input and streptavidin beads. While these parameters may vary depending on factors such as the PL enzyme used, the expression level of the PL construct, and the cell type, our study provides a guide for parameter optimization. In addition, we evaluated different binding times for the biotinylated proteins to streptavidin beads. While an 18-h incubation exhibited the best performance in terms of protein identification, shortening the binding time to 4 h resulted in only a marginal decrease in protein identification, suggesting the possibility of shortening the protocol to one day. Furthermore, in addition to improving reproducibility in sample preparation, we adopted a DIA-based MS method for the measurement of biotinylated proteins, offering enhanced reproducibility and consistency in protein identification and quantification (Fig. 3). The consistency in quantification is of particular importance for studies across multiple conditions, for example, receptor activation over a time course. In previous APEX2 studies, we used a combination of DDA-based and targeted proteomics approaches to ensure consistent quantification (Lobingier et al, 2017; Polacco et al, 2024), which duplicated the measurement time compared to using the DIA approach itself. Integration of manual enrichment with DIA-based protein quantification already improved the reproducibility in protein identification from 78.7 to 95% overlap among three replicates. The incorporation of automated enrichment not only boosted throughput but also reduced variability in protein quantification from a median CV of 13% to 7%. While our strategy automated the enrichment of biotinylated proteins, including binding to streptavidin beads followed by rigorous washing steps, on the KingFisher Flex platform, subsequent proteolytic digestion of biotinylated proteins and clean-up for MS analysis were performed manually in 96-well plate format. Several recent studies have proposed methodologies to automate these additional steps (Fu et al, 2018; Liu et al, 2021; Müller et al, 2020), promising further reduction in hands-on time for sample preparation and increased reproducibility, albeit requiring an additional liquid handling platform.

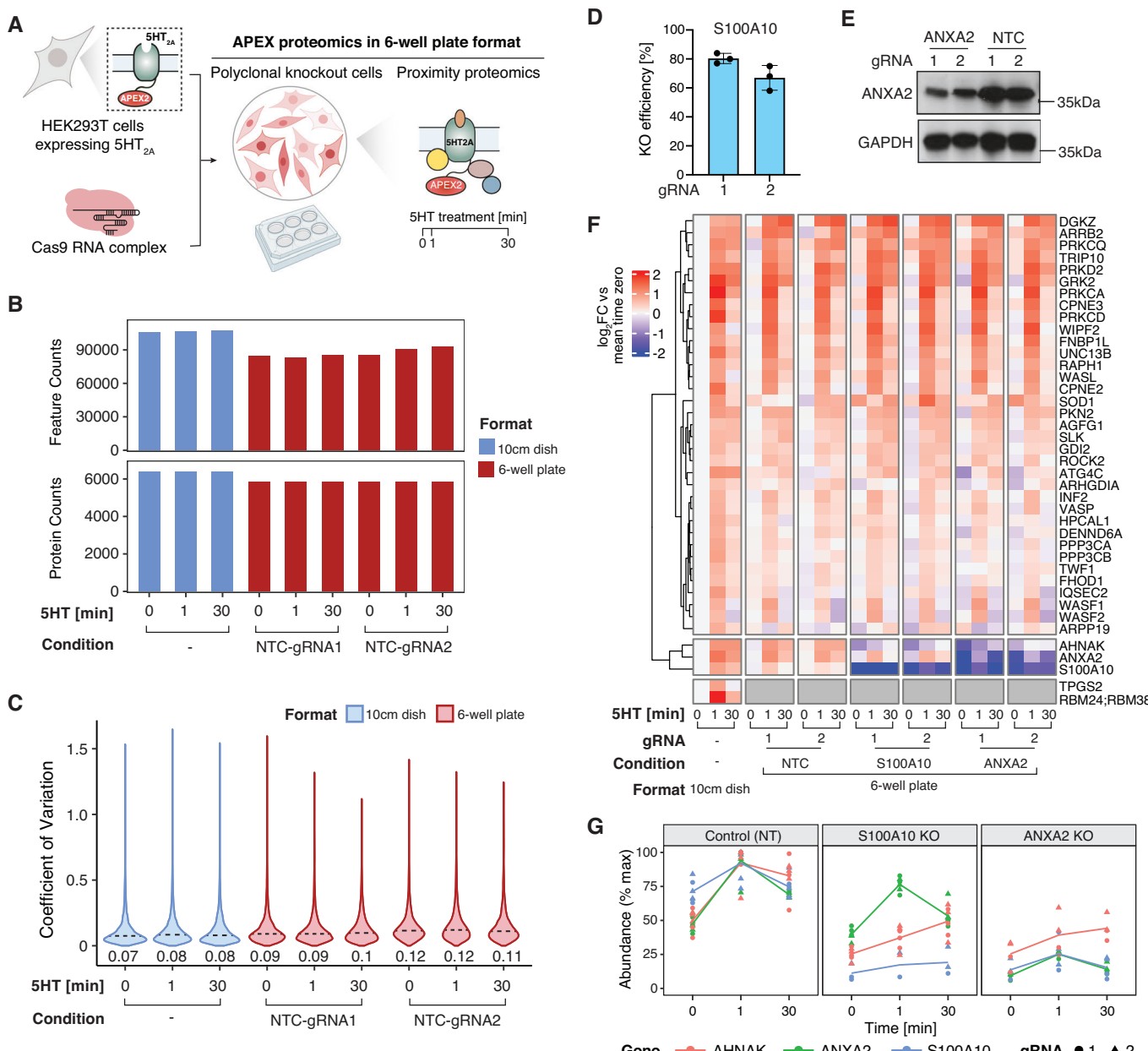

**Figure 6.  Exploring proximal interaction network perturbations induced by CRISPR-based gene knockout using low-input PL pipeline.**

(**A**) Schematic of PL with CRISPR-based gene knockout (KO). Polyclonal knockout cell lines were generated by electroporation of Cas9 ribonucleoprotein complexes into HEK293T cells expressing the APEX2-fused 5HT$_{2A}$ receptor. The KO cells were seeded in a 6-well plate for proximity proteomics with receptor activation by 5HT for 0, 1, and 30 min. (**B**) Performance comparison of 5HT$_{2A}$ PL in 10 cm dish and 6-well plate format based on feature and protein identification. Two independent non-targeting controls (NTC) were used for the 6-well plate format. All numbers in the figures represent the sum of identifications of feature and protein level across three biological replicates, $n = 3$. (**C**) Distribution of the coefficient of variation (CV) of protein intensities from three replicates comparing 5HT$_{2A}$ PL in 10 cm dish and 6-well plate format. Two independent non-targeting controls (NTC) were used for the 6-well plate format. Numbers depicted on the graph represent median CVs. Data were collected from three independent biological replicates, $n = 3$. (**D**) Knockout validation for S100A10 by Inference of CRISPR Edits (ICE), $n = 3$ biological replicates. Results are presented as mean ± standard error of the mean (SEM). (**E**) Knockout validation for ANXA2 by Western blot analysis. (**F**) Heatmap depicting proteins from 5HT2A-APEX2 experiment with transient responses to 5HT treatment (Fig. 5D) compared across NTCs and knockouts of S100A2 and ANXA2 in 6-well plate format and the 5HT2A APEX2 data from 10 cm dishes. Data were collected from three independent biological replicates ($n = 3$) for NTCs and S100A10 and two independent biological replicates ($n = 2$) for ANXA2. (**G**) Temporal profile for regulated proteins in 5HT$_{2A}$ receptor proximal interaction network in KO vs. control (non-targeting) cells. Line charts represent the log2 fold change over the time course of receptor activation with 0.1 µM 5HT. Red: protein AHNAK, Green: protein ANXA2, Blue: protein S100A10, Solid circle: KO using gRNA 1, Solid triangle: KO using gRNA 2. Data were collected from three independent biological replicates ($n = 3$) for NTCs and S100A10 and two independent biological replicates ($n = 2$) for ANXA2.

We showcased the versatility of our automated proximity proteomics approach in mapping subcellular proteomes by targeting the APEX2 PL enzyme to early and late endosomes, the Golgi apparatus, lysosomes, and plasma membrane (Fig. 4). The inter-compartmental crosstalk and membrane contact sites pose challenges in deciphering compartment-specific proteins solely based on existing knowledge (Go et al, 2021; Eisenberg-Bord et al, 2016). Hence, we used the PL proteome generated by cytosolic APEX2-construct (NES-APEX2) as control and required proteins that were significantly overrepresented in the subcellular compartments compared to the control to be exclusively present in one compartment. Using these stringent filtering criteria, we identified 337, 55, and 228 location-specific proteins for endosome and lysosome/late endosome, Golgi, and plasma membrane, respectively. The majority of our compartment-specific proteins aligned with their location defined in the Human Cell Map (Go et al, 2021), a proximity map of the HEK293 proteome generated using PL. Discrepancies between our dataset and the Human Cell Map could be attributed to the usage of different proximity labeling enzymes (APEX2 versus BioID), different proteins or protein localization domains employed to target the enzyme to a subcellular location, as well as variances in data analysis criteria. Notably, expression of an APEX2 construct (or other proximity labeling constructs) at a subcellular compartment could potentially impact the function of the compartment. We envision that our scalable proximity labeling strategy will facilitate the investigation of dynamics in subcellular protein localization across various experimental conditions.

As a second application of the automated proximity proteomics strategy, we employed it to investigate temporal changes in protein interactions. Previously, we demonstrated that APEX2-based PL could delineate protein interaction networks of GPCRs and their agonist-dependent changes with temporal and spatial precision (Lobingier et al, 2017; Paek et al, 2017). To ensure consistent and reproducible quantification over the time course, we used a two-step quantitative MS workflow consisting of DDA for identification followed by targeted proteomics for quantification of biotinylated proteins in our prior work (Lobingier et al, 2017). Here, we demonstrated the capability of our automated PL approach combined with DIA-MS to yield high reproducibility in quantifying agonist-induced proximal proteomes over a time course. We delineated the changes in the proximal protein interaction network of $5HT_{2A}$ upon activation with its endogenous ligand, 5HT (Fig. 5). The 744 proteins exhibiting significant changes in the $5HT_{2A}$ PL dataset displayed a down-regulation of plasma membrane proteins and an up-regulation of endosomal proteins, indicative of receptor internalization and trafficking following activation. To account for location-specific proteins that reside in the local environment of $5HT_{2A}$ and enrich for functionally relevant proteins in the receptor's proximal interaction network, we filtered the dataset for significant proteins using a defined fold change cutoff. This filtering criteria retained proteins with well-characterized functions in GPCR signaling and trafficking, such as GRK2, ARRB2, PKC, and PKD. Notably, the transient nature of these interactions has traditionally posed challenges for their capture using conventional affinity purification coupled to MS. Furthermore, although HEK293T cells serve as an excellent model system for demonstrating the efficacy of the automated PL approach to detect PPI dynamics with high sensitivity, these cells do not endogenously express the $5HT_{2A}$ receptor. Therefore, we emphasize the necessity of conducting PPI mapping in other cell types expressing endogenous $5HT_{2A}$, ideally by endogenously tagging the receptor with the APEX enzyme to avoid artifacts from overexpression, and/or conducting functional characterization to validate any hypotheses derived from the proximal interaction networks. Functional studies could for example include perturbation of the suggested proximal interactors followed by assays monitoring downstream signaling effects, such as intracellular calcium, or receptor trafficking.

Finally, we assessed the feasibility of using our PL pipeline for monitoring proximal interaction network perturbations induced by CRISPR-based gene knockout (KO) (Fig. 6). To achieve this, the PL pipeline was modified to accommodate lower sample inputs, which compared to the high-input PL pipeline maintained similar performance based on protein identification and quantitative consistency. Although the sample input amounts were scaled down for the 6-well plate cell culture format, the increasing sensitivity of MS instrumentation suggests that further reduction to the 12- or even 24-well plate format, with adjustments to parameters such as buffer and bead volumes, would be feasible. Following protocol downscaling, we examined the impact of CRISPR-based KO of individual network components on the activity-dependent $5HT_{2A}$ interaction network. Specifically, ANXA2 and S100A10, two proteins in the proximal interaction network of $5HT_{2A}$, were selected for KO, with the effects assessed using the low-input PL pipeline. In ANXA2-KO cells, a significant reduction in S100A10 protein levels was observed. This is consistent with prior studies showing reduced protein levels of p11 in various tissues from Anxa2-KO mice (He et al, 2008), suggesting a role for ANXA2 interaction in the stabilization of S100A10 (P11). The converse knockout of S100A10 in our study did not affect ANXA2 levels. Furthermore, both S100A10-KO and ANXA2-KO cells showed reduced labeling of the protein AHNAK across all conditions. This observation also aligns with previous studies demonstrating AHNAK's interaction with the ANXA2-S100A10 complex (Jin et al, 2020), and the requirement of ANXA2 and S100A10 for its recruitment to the plasma membrane (Benaud et al, 2004). Notably, investigation into the effect of ANXA2 and S100A10 KO on $5HT_{2A}$ receptor, specifically receptor expression at the plasma membrane, revealed a slight decrease in both KO cell lines. Though the interaction between the S100A10/ANXA2 heterotetramer and AHNAK has been well-characterized (Benaud et al, 2004; Jin et al, 2020; Chen et al, 2022), it is the first time to suggest their potential participation in the $5HT_{2A}$ interaction network. The S100A10/ANXA2 heterotetramer interacts with a consensus binding motif in the C-terminal region of AHNAK (Chen et al, 2022; Jin et al, 2020). AHNAK consists of a PDZ domain in the N terminal region, so possibly the binding of AHNAK/S100A10/ ANXA2 is mediated by the PDZ binding motif in 5-HT2A. Other proteins in the 5HT2A proximal interaction network, aside from AHNAK, S100A10, and ANXA2 are not affected by the KO. This is consistent with the hypothesis that the three proteins form a complex to help receptor trafficking to the cell surface, while the receptor at the cell surface remains functional. However, further functional validation is needed to confirm our findings and their exact mechanism. Overall, the results highlight the efficacy of the automated, low-input PL pipeline in conjunction with CRISPR

KO of proximal interactors for delivering reproducible results and monitoring interaction network perturbations across multiple conditions.

Collectively, we have developed a PL proteomics approach enabling the mapping of subcellular proteomes and the characterization of spatially and temporally resolved proximal interaction networks with high throughput and reproducible quantification. The method presented here holds great promise for large-scale proximity labeling studies. While we demonstrated this automated proximity proteomics for APEX2-based PL and implemented it on a KingFisher Flex platform, we anticipate its broad applicability in two aspects: (1) its potential extension to various other PL enzymes that function by biotinylating proteins, and (2) its seamless adaption to other automated platforms capable of handling magnetic beads.

# Methods

### Reagents and tools table

| Reagent/resource | Reference or source | Identifier or catalog number |
| --- | --- | --- |
| **Cell lines** | | |
| HEK293T/17 cells | ATCC | CRL-11268 |
| **Antibodies** | | |
| Mouse anti-FLAG (M1) | Sigma | F-3040 |
| Mouse anti-Vimentin | Invitrogen | MA3-745 |
| Rabbit anti-E Cadherin | CST | 3195T |
| Mouse anti-RAB5A | Fisher | 89333555 |
| Mouse anti-RAB9 | Invitrogen | MA3-067 |
| Rabbit anti-Golgin-97 | CST | 13192 |
| Chicken anti-GFP | VWR | RL600-901-215 |
| AF488-labeled goat anti-Chicken | Invitrogen | A11039 |
| AF555-labeled goat anti-Mouse | Invitrogen | A21422 |
| AF647-labeled goat anti-Rabbit | Invitrogen | A21244 |
| Rabbit HRP anti-GAPDH | Bio-Legend | 607904 |
| HRP anti-Streptavidin | VWR | N100 |
| **Recombinant DNA** | | |
| NES-APEX2 | This study | N/A |
| 2xFYVE-APEX2 | This study | N/A |
| GalT-APEX2 | This study | N/A |
| LAMP1-APEX2 | This study | N/A |
| LAMTOR1-APEX2 | This study | N/A |
| Lyn11-APEX2 | This study | N/A |
| 5HT$_{2A}$-APEX2 | This study | N/A |
| G$\alpha_q$-RLuc8 | Addgene | 140982 |
| G$\beta_3$ | Addgene | 140988 |
| GFP2-G$\gamma_9$ | Addgene | 140991 |
| 5HT$_{2A}$ | (Kim et al, 2020) | N/A |

| Reagent/resource | Reference or source | Identifier or catalog number |
| --- | --- | --- |
| **Chemical, enzymes, and other reagents** | | |
| Dulbecco's modification of Eagle's medium (DMEM) | Corning | 10-013-CV |
| Fetal bovine serum (FBS) | Gibco | A31605-01 |
| Dulbecco's Phosphate-Buffered Saline (DPBS) | Corning | 21-031-CV |
| DPBS, calcium, magnesium | Gibco | 14040182 |
| 0.05% Trypsin-EDTA | Gibco | 25300-054 |
| Penicillin streptomycin solution | Corning | 30-002-CI |
| PolyJet | SignaGen | SL100688 |
| 0.45 µm PVDF filter unit | Millipore | SE1M003M00 |
| Cellstripper | Corning | 25-056-CI |
| Attune performance tracking beads | Thermo Fisher | 4449754 |
| ECL western blotting substrate | Pierce | 32106 |
| Streptavidin magnetic beads | Pierce | 88817 |
| Biotin phenol (Biotin Tyramide) | Iris Biotech | LS-3500 |
| Hydrogen peroxide | Sigma | H1009-100ML |
| Sodium azide | Sigma | S2002 |
| Sodium ascorbate | Spectrum | S1349 |
| Trolox (6-hydroxy-2,5,7,8-tetramethylchromane-2-carboxylic acid) | Sigma | 238813 |
| 16% Formaldehyde (w/v) | Pierce | 28908 |
| EverBrite mounting medium with DAPI | Biotium | 23002 |
| cOmplete protease inhibitor cocktail tablets mini, EDTA-free | Roche | 11846170001 |
| 660 nm protein assay kit | Pierce | 22662 |
| TCEP | Pierce | 20490 |
| Dithiothreitol (DTT) | Sigma | D0632 |
| Iodoacetamide (IAA) | Sigma | I1149 |
| Sequencing-grade modified trypsin | Promega | V5111 |
| Lysyl endopeptidase | VWR | 100369-822 |
| BioPureSPE 96-well C18 plate | NEST | HNS S18V |
| Ascorbic acid | Sigma | A5960 |
| BSA, free fatty acid | Akron | AK8909 |
| Coelenterazine 400a | Nanolight | 340-1 |
| Dialyzed FBS | Gibco | A3382001 |
| Hank's Balanced Salt Solution | Gibco | 14065056 |
| Sodium chloride | Sigma | S7653-250G |
| Tris HCl pH 7.5 | Corning | 46-030-CM |
| Tris HCl pH 8.0 | Corning | 46-031-CM |

| Reagent/resource | Reference or source | Identifier or catalog number |
|---|---|---|
| Sodium deoxycholate | Sigma | D6750-500G |
| Triton-x100 | Sigma | T9284-100ML |
| Sodium dodecyl sulfate | Fisher BioReagents | BP166-500 |
| Urea | Promega | V3171 |
| Potassium chloride | Sigma | P9541-1KG |
| Sodium carbonate | Sigma | S2127-500G |
| 5-hydroxytryptamine hydrochloride | TCI | S0370 |
| Pierce 660 nm Protein Assay Reagent | Thermo Scientific | 22660 |
| SF cell line 96-well nucleofector kit | Lonza Walkersville | V4SC2096 |
| HEPES buffer | Corning | 25-060-CI |
| **Software** | | |
| Spectronaut (version 16.0) | Biognosys | https://biognosys.com/shop/spectronaut |
| MSstats (version 4.4.0) | Bioconductor | https://www.bioconductor.org/packages/release/bioc/html/MSstats.html |
| Attune NxT software | Thermo Fisher Scientific | |
| FlowJo | FlowJo, LLC | https://www.flowjo.com/ |
| Fiji | imageJ | https://imagej.net/software/fiji/ |
| NIS-Elements software (v. 5.30.01 build 1541) | Nikon | |
| Prism (v8.0) | GraphPad Software | |
| **Other** | | |
| Orbitrap Exploris 480 MS with internal calibration option | Thermo Fisher Scientific | BRE725533 |
| EASY-nLC 1200 system | Thermo Fisher Scientific | |
| Nikon Ti2-E microscope | Nikon | |
| Attune NxT Flow Cytometer | Thermo Fisher Scientific | |
| KingFisher Flex system | Thermo Fisher Scientific | |
| 4D-Nucleofector | Lonza Bioscience | |

## Methods and protocols

### Mammalian cell culture conditions

HEK293T cells (CRL-1583, ATCC) were maintained in Dulbecco's modified Eagle's medium (DMEM, GIBCO or Fisher Scientific) supplemented with 10% (v/v) fetal bovine serum (FBS) (GIBCO) and 1% penicillin/streptomycin solution at 37 °C in a 5% $CO_2$ humidified incubator.

### cDNA constructs

Standard cloning techniques were used to generate all constructs. Vectors were digested using restriction enzymes (EcoRI and BamHI) and PCR fragments were amplified using Q5 DNA polymerase (NEB). NES-APEX2, Lyn11-APEX2, 2xFYVE-APEX2, LAMTOR1-APEX2, LAMP1-APEX2, GalT-APEX2, and $5HT_{2A}$-APEX2 were cloned into pLVX-TetOne-Puro by in-Fusion HD cloning. Ligated plasmid products were transformed into stellar competent cells (*E. coli* HST08 strain, Takara Bio). See Dataset EV1 for full descriptions of constructs.

### Lentiviral generation and transductions

To generate lentiviruses, $8 \times 10^6$ HEK293T cells were seeded in T175 flask. The next day, cells were transfected with 5 µg of each of the pLVX-TetOne-Puro lentiviral plasmids containing the gene of interest, 3.33 µg of Gag-Pol-Tat-Rev packaging vector, and 1.66 µg of VSV-G envelope vector with 30 µL PolyJet reagent (3 µL per µg DNA) in 250 µL serum-free DMEM (25 µL per µg DNA). After 24 h, the culture media was replaced with fresh DMEM supplemented with 10% FBS and 1% penicillin-streptomycin solution. Virus was harvested at 48 h post media change. Briefly, the media was collected and centrifuged at 2500 rpm for 5 min. The supernatant was filtered through a 0.45-µm PVDF filter and mixed with the PEG-salt solution (a final 8.5% PEG and 0.3 M NaCl). The virions were aggregated and precipitated at 4 °C for at least 2 h, followed by centrifugation at 3500 rpm for 20 min. Immediately after spinning, the virus pellet was resuspended with 500 µL sterile PBS. For generation of stable APEX2-tagged cell lines, HEK293T cells were infected with lentivirus at ~50% confluency and selected by 2 µg/mL puromycin in growth medium.

### Flow cytometric analysis

To create monoclonal stable cell lines, polyclonal cells were single-cell sorted in a 96-well plate by fluorescence-activated cell sorting (FACS) using BD FACS Aria or BD FACS Fusion. Polyclonal cells were washed once with DPBS, collected by using Cellstripper (Corning). In the case of non-GFP tagged expression construct ($5HT_{2A}$-APEX2), cells were incubated with Alexa 647-conjugated M1-anti-FLAG antibody (1:1000 dilution) at 4 °C for 40 min. After spinning down, cells were washed with ice-cold PBS and resuspended in 1 mL of sample buffer (PBS/1 mM EDTA/25 mM HEPES/1% FBS) for sorting. 24 clones were selected and expanded in a 24-well plate for characterization by flow cytometry using the Attune NxT Flow Cytometer. The cellular morphology was first gated based on an FSC-A vs. SSC-A (forward scatter - area vs. side scatter - area) plot. Next, gating was generated based on the FSC-A vs. FSC-H (area vs. height) plot to select singlets. These singlets were further analyzed to identify the cells expressing $5HT_{2A}$ or the localization domains based on Alexa 647 or GFP fluorescence signal, respectively. The data was processed in FlowJo.

To quantify the $5HT_{2A}$ receptor at the plasma membrane, the non-targeting controls (NTC), ANXA2 KO, and S100A10 KO cells were washed with an ice-cold sample buffer (PBS/1 mM EDTA/25 mM HEPES/1% dialyzed FBS) and lifted in Cellstripper for 5 min at room temperature. After neutralization with an equal volume of cell culture media (DMEM/10% dialyzed FBS/1%

penicillin/streptomycin), the cells were collected and incubated with Alexa 647-conjugated M1-anti-Flag antibody (1:1000 dilution) at 4 °C for 40 min. Median fluorescence intensity of 10,000 cells per condition was measured using Attune NxT Flow Cytometer.

### Confocal fluorescence imaging

HEK293T cells expressing NES-APEX2, Lyn11-APEX2, 2xFYVE-APEX2, LAMTOR1-APEX2, LAMP1-APEX2, and GalT-APEX2 were plated onto 12 mm collagen-coated glass coverslips in 12-well plate and treated with 1 µg/mL doxycycline for 24 h. For initial polyclonal and monoclonal cell lines selection, after aspirating culture media, cells were fixed with 4% paraformaldehyde in PBS at room temperature (RT) for 15 min. Cells were then mounted onto glass slides with EverBrite mounting medium with DAPI (Biotium Cat #23002), and ready for imaging. For the final monoclonal cell line characterization, the localization of all spatial APEX2 constructs were determined by the co-localization analysis with subcellular location markers. Cells were first washed with PBS twice and fixed in 4% paraformaldehyde in PBS for 15 min at room temperature (RT). After washing with PBS, cells were permeabilized with 0.2% Triton X-100 in PBS for 10 min at RT and blocked in 5% BSA (w/v) in PBS with 0.1% Tween-20 (PBS-T) for 1 h at RT. Incubation with primary antibodies diluted in 1% BSA in PBS-T against GFP (1:500) and Vimentin (1:100) for NES, GFP (1:500) and E-cadherin (1:500) for Lyn11, Rab 5 (1:200) for 2xFYVE, Rab9 (1:200) for LAMTOR1 and LAMP1, Golgin-97 (1:200) for GalT, and Alexa 647-conjugated M1-anti-FLAG antibody (1:200) for 5HT$_{2A}$ was performed at 4 °C overnight. After washing with PBS-T three times, cells were incubated with AF488-labeled goat anti-Chicken, AF555-labeled goat anti-Mouse, and AF647-labeled goat anti-Rabbit secondary antibodies diluted in 1% BSA in PBS-T (1:1000) for 1 h at RT. Cells were then washed in PBS-T twice and one final wash in PBS, mounted onto glass slides using EverBrite mounting medium with DAPI (Biotium Cat #23002), and imaged by confocal fluorescence microscopy. Images were generated using a Nikon Ti2-E microscope equipped with a Crest X-Light-V2 spinning disk confocal (Crest Optics), emission filters, 438/24 (DAPI), 511/20 (GFP), 560/25 (RFP), and 685/40 (Cy5), and Celeste Light Engine excitation lasers 405/477/546/638 nm used respectively (Lumencor), Piezo stage (Mad City Labs), and a Prime 95B 25 mm CMOS camera (Photometrics) using a Plan Apo VC 100x/1.4 Oil (Nikon). The data was captured with NIS-Elements software (v. 5.41.01 build 1709, Nikon) and processed in Fiji/ImageJ2 (Schindelin et al, 2012).

### BRET assay

Bioluminescence resonance energy transfer (BRET) assays were performed to validate the functionality of the 5HT$_{2A}$-APEX2 construct. Here, the TRUPATH platform for measuring G protein dissociation was used as previously described (Olsen et al, 2020; Kim et al, 2020; DiBerto et al, 2022). In brief, wild type or APEX2 constructs of 5-HT$_{2A}$ were co-transfected with Ga$_q$-RLuc8, Gb$_3$, and GFP2-Gg$_9$ in a 1:1:1:1 ratio in HEK293T cells maintained in Dulbecco's Modified Eagle Media (DMEM) supplemented with 10% fetal bovine serum (FBS) and 1% penicillin-streptomycin (pen-strep). After 8 h, the media were exchanged for DMEM supplemented with 1% dialyzed FBS and 1% pen-strep to minimize receptor desensitization by serum serotonin, and at least 12 h before the experiment, cells were plated in 96-well microplates in DMEM supplemented with 1% dialyzed FBS and 1% pen-strep. To conduct the experiments, plates were first vacuum aspirated and

60 mL of assay buffer (1x Hank's Balanced Salt Solution in phosphate-buffered saline, 20 mM HEPES, pH 7.4) were added to the wells. Next, 10 mL of coelenterazine 400a diluted in the assay buffer (50 mM) were added to the wells, and allowed to incubate for 5 min. Next, 30 mL of 3x drug diluted in drug buffer (assay buffer supplemented with 3 mg/mL fatty acid-free bovine serum albumin and 0.3 mg/mL ascorbic acid) were added to the wells and allowed to incubate for 5 min. Last, plates were read using a BMG Labtech PHERAstar FSX with BRET$^2$ plus optic module. All data were analyzed using GraphPad Prism (V8.0).

### APEX proximity labeling

HEK293T cells stably expressing 5HT$_{2A}$-APEX2 and protein localization domains (NES-APEX2, Lyn11-APEX2, 2xFYVE-APEX2, LAMTOR1-APEX2, LAMP1-APEX2, and GalT-APEX2) were seeded with 350 K cells/well in 6-well plate for testing biotinylation by western blot, and 3.5 × 10$^6$ cells/dish in 10 cm dish for enrichment of biotinylated proteins followed by quantitative MS. Following 24 h doxycycline induction, cells were incubated with 500 µM biotin phenol at 37 °C for 30 min in complete medium (DMEM/10% FBS/1% pen/strep), but 5HT$_{2A}$-APEX2 cells were seeded with 10% dialyzed FBS/DMEM/1% pen/strep and changed to 1% dialyzed FBS/DMEM/1% pen/strep during doxycycline induction. Meanwhile, 0.1 µM 5HT was added for incubation of 1, 5, 10, and 30 min in cells expressing 5HT$_{2A}$-APEX2 to capture the protein interaction dynamics. There was no agonist incubation for all APEX2-tagged localization domains. 2 mM H$_2$O$_2$ diluted in complete medium was freshly prepared prior to use. The H$_2$O$_2$ containing media was mixed with biotin phenol-containing media at 1:1 ratio (v/v) to initiate APEX labeling. The labeling reaction was quenched after 45 sec by removing the medium, washing cells three times with ice-cold quenching solution (PBS supplemented with 10 mM sodium ascorbate, 5 mM Trolox, and 10 mM sodium azide). Cells were collected in a quenching solution (1 mL for 6-well plate and 8 mL for 10 cm dish) and pelleted by centrifugation at 3000 × g for 10 min at 4 °C. The supernatant was removed and the cell pellet was resuspended in lysis buffer (50 mM Tris, 150 mM NaCl, 0.5% sodium deoxycholate, 0.1% SDS, 1% Triton X-100, 10 mM sodium ascorbate, 5 mM Trolox, and 10 mM sodium azide, 1 mM DTT, and cOmplete protease inhibitor) supplemented with 10 mM sodium ascorbate, 5 mM Trolox, and 10 mM sodium azide, 1 mM DTT, and protease inhibitor (100 µL for small-scale analysis and 1 mL for large-scale analysis). With a freeze-thaw cycle, each sample was sonicated for 5 s twice, centrifuged at 13,000 × g for 10 min at 4 °C. The supernatant was subjected to the automated enrichment of biotinylated proteins. The manual enrichment was performed in parallel with the automated enrichment protocol, but using a magnetic rack (DynaMag-2, Thermo Fisher). The protein concentration was determined using Protein assay 660 (Pierce). For each proteomic sample, 25 µL lysate before and after binding was saved for western blot analysis to evaluate biotinylation and enrichment efficiency of biotinylated proteins.

### Western blot analysis

Cell lysates were mixed with equal volume of sample loading buffer (200 mM β-mercaptoethanol in NuPAGE LDS sample buffer), boiled at 95 °C for 20 min, separated on 4–12% SDS-PAGE gels (BioRad), and transferred to PVDF membranes (Trans-blot turbo transfer system, BioRad). The blots were blocked in 5% milk in

TBS-T (Tris-buffered saline with 0.1% Tween 20) at 4 °C overnight and washed three times with TBS-T for 10 min. Blots were incubated with horseradish peroxidase (HRP)-conjugated anti-Streptavidin antibody (1:5000 dilution using 2% BSA in TBS-T) or HRP-conjugated anti-GAPDH antibody (1:10,000 dilution) at room temperature for 1 h. After washing three times in TBS-T for 10 min, the blots were incubated with ECL western blotting substrate (Pierce) for 1 min. Chemiluminescent signals were captured on the Azure 400 (Azure biosystem).

### Automated enrichment protocol for biotinylated proteins

For the automated biotinylation enrichment protocol, the Kingfisher Flex system (Thermo Fisher) is programmed to simultaneously process a maximum of 96 samples. This protocol below includes two parts, where part 1 (Plate 1–3) is for washing magnetic streptavidin beads and binding of the biotinylated proteins to beads, and part 2 (Plate 4–12) is for washing and collecting beads prior to Lys-C/trypsin digestion. The enrichment protocol is conducted in the cold room using deep-well plates. See Dataset EV2 for full description of the Kingfisher program. A detailed step-by-step protocol is provided on protocols.io (https://doi.org/10.17504/protocols.io.yxmvm3jbbl3p/v1).

Plate 1. Add 100 µL of 50% streptavidin magnetic bead slurry (Pierce) to each well of a 96-well plate and add 900 µL RIPA buffer (50 mM Tris, 150 mM NaCl, 0.5% sodium deoxycholate, 0.1% SDS, 1% Triton X-100, pH 7.4) to each well. The beads are mixed for 2 s and slowly washed for 30 s. At the end of this step, the beads are collected for 5 counts with collection time of 10 s/count. A total of 4 min for washing.

Plate 2. Add 1 mL RIPA buffer. The beads are transferred to this plate and slowly washed for 4 min.

Plate 3. Add 1 mg protein from cell lysate to each well and top-up to 1 mL with RIPA buffer. The beads are transferred to this plate for overnight binding (~12 h).

Plate 4–6. Add 1 mL RIPA buffer to each plate. The beads with bound biotinylated proteins are collected from Plate 3 and transferred to Plate 4–6 for washing three times with RIPA buffer. Each wash step takes 7 min.

Plate 7. Add 1 mL 1 M KCl solution. Wash the beads for 4 min.

Plate 8. Add 1 mL 0.1 M $Na_2CO_3$ solution. Wash the beads for 4 min.

Plate 9. Add 1 mL freshly prepared 2 M urea in 50 mM Tris-HCl (pH 8.0). Wash the beads for 4 min.

Plate 10–11. Add 1 mL 50 mM Tris-HCl (pH 8.0). Wash the beads twice for 5 min.

Plate 12. Add 200 µL of freshly prepared 2 M urea in 50 mM Tris-HCl buffer (pH 8.0). The beads with bound biotinylated proteins are collected in this plate for on-bead digestion.

### Sample preparation for MS analysis

Samples were reduced with 5 mM TCEP at 37 °C for 30 min by shaking at 1000 rpm, followed by alkylation with 5 mM iodoacetamide (IAA) at room temperature (RT) for 30 min. Extra IAA was quenched by 5 mM dithiothreitol (DTT) for 10 min at RT. Samples were then digested with 1 µg Lys-C and 1 µg trypsin at 37 °C for 6 h and 25 °C for 13 h. 0.5 µg trypsin was added to each sample for an additional 2 h incubation at 37 °C. Following digestion, the supernatant was transferred to a new 96-well plate and 10 µL of 10% trifluoroacetic acid (TFA) was added to each sample to a final

pH 2–3. The peptide samples were desalted using C18 96-well plate (BioPureSPE, HNS S18V-20mg, the Nest group) by centrifugation at 1600 rpm for 2 min at each step. The plate was activated with 100 µL methanol, washed three times with 100 µL 80% acetonitrile (ACN)/0.1% TFA, and equilibrated three times with 100 µL 2% ACN/0.1% TFA. After loading samples, the plate was washed three times with 100 µL 2% ACN/0.1% TFA and eluted twice with 55 µL 50% ACN/0.25% formic acid (FA). Samples were dried by vacuum centrifugation, resuspended in 20 µL 0.1% formic acid, and injected 1 µL for MS analysis.

### Mass spectrometric data acquisition

Digested peptide mixtures were analyzed on an Orbitrap Exploris 480 MS system equipped with an Easy nLC 1200 ultra-high pressure liquid chromatography system. Samples were injected on a C18 reverse phase column (15 cm × 75 µm I.D. packed with BEH C18 1.7 µm particles, Waters) in 0.1% formic acid (FA). Mobile phase A consisted of 0.1% FA and mobile phase B consisted of 80% acetonitrile (ACN)/0.1% FA. Peptide mixtures were separated by mobile phase B ranging from 4% to 16% over 40 min, followed by an increase to 28% B over 26 min and 44% B over 4 min, then held at 95% B for 10 min at a flow rate of 300 nL/min.

To generate a spectral library, one of three biological replicates in each experiment was acquired using data-dependent acquisition (DDA). DDA analysis consisted of one full scan over a $m/z$ range of 350–1050 in the Orbitrap at a resolving power (RP) of 120 K with an RF lens of 40%, a normalized automatic gain control (AGC) target of 300%. The 20 most intense precursors at charge states of 2–6 from the full scan were selected for higher energy collisional dissociation (HCD) FTMS[2] analysis at RP 15 K with an isolation width of 1.6 Da, a normalized collision energy (NCE) of 30, an AGC target of 200%, and a maximum injection time of 22 ms. Dynamic exclusion was enabled for 30 s with a repeat count of 1.

The parameters evaluated for data-independent acquisition (DIA) method include mass scan range of 350–1050 and 390–1010, MS[2] scans with a normalized HCD (higher-energy collisional dissociation) collision energy of 27%, 30%, and 33%, maximum injection time of 22 ms, 32 ms, 42 ms, dynamic 6-point, dynamic 8-point, and Auto, and window overlap width of 0, 1, and 2 $m/z$. In the end, all samples were analyzed using DIA-MS by collecting one full scan over a range of 350–1050 $m/z$ in the Orbitrap at RP 120 K with an AGC target of 300% and an automatic maximum injection time, followed by DIA MS[2] scans over 350–1050 $m/z$ at RP 15 K using an isolation window of 20 $m/z$, an overlapping isolation window of 2 $m/z$, a normalized HCD collision energy of 30%, an AGC target of 200%, and a maximum injection time of 32 ms.

### Protein identification and quantification

The DDA raw data for automated enrichment combined with data-dependent acquisition MS (Auto_DDA) were analyzed using MaxQuant (version 1.6.12.0) for identification of peptides and proteins. The DIA raw data for manual enrichment protocol combined with data-independent acquisition MS (Manual_DIA), automated enrichment protocol with data-independent analysis (Auto_DIA), and DIA-MS method optimization were analyzed using Spectronaut (version 16.0) with direct DIA search. All the raw files were searched against the SwissProt Human database (downloaded 10/2020) using default settings with variable

modification of methionine oxidation and protein N-termini acetylation, and fixed modification of cysteine carbamidomethylation. The data were filtered to obtain a false discovery rate of 1% at the peptide spectrum match and the protein level.

In the data analysis of methods optimization, the DIA datasets were searched against the SwissProt Human database (downloaded 10/2020) by using a spectral library-free approach (direct-DIA) in Spectronaut (version 16.0). We observed a slightly higher consistency in precursor identifications across runs when comparing library-free and library-based analysis of subcellular proteomes, so in the data analysis of mapping subcellular proteomes or protein interaction dynamics, the DDA datasets were collected to construct a spectral library. All the raw data files were searched against the SwissProt Human database (downloaded 10/2020) by using the Pulsar search engine integrated into Spectronaut (version 16.0). The raw mass spectra files were processed using the default BGS settings with digestion enzyme of trypsin, fixed modification of cysteine carbamidomethylation, variable modification of methionine oxidation and protein N-termini acetylation and filtered to a 1% false discovery rate (FDR) at the peptide spectrum match (PSM), peptide, and protein level. The DIA datasets were searched against the generated DDA spectral libraries in Spectronaut with default BGS setting, but the cross-run normalization was disabled.

### Statistical analysis of proteomic data and data visualization

Proteomic statistical analyses were completed using the R programming language, version 4.2.0. Peptide ion intensities from Spectronaut were summarized to protein intensities using the package MSstats (version 4.4.0) and its function dataProcess enabling options to turn off model-based imputation and turning on removal of uninformative features and outliers (Tsai et al, 2020). All downstream analyses were performed on log2 transformed protein intensities.

Subcellular proteomes were compared for each protein or protein group using the built-in R function lm to estimate additive effects for both subcellular location and replicate batch, and the emmeans function from package emmeans (version 1.7.3) to calculate pairwise subcellular location $\log_2$ fold changes and t-test statistics including $p$ values. Adjusted $p$ values were calculated as false discovery rates using the Benjamini–Hochberg method available in R built-in function p.adjust. Selection of location-specific proteins was made by identifying proteins with significant labeling increases when compared to the cytosolic NES-APEX2 sample ($\log_2 FC > 1$, unadjusted p.value < 0.005), and requiring the increase to be exclusive to GalT, Lyn11, or to the combined 2xFYVE, LAMTOR1, LAMP1 set. Gene ontology enrichment analysis of location-specific protein sets was performed using the enricher function from package clusterProfiler (version 4.4.1) and human gene ontology annotations in the Bioconductor annotation package org.Hs.eg.db (version 3.15.0). The Human Cell Map data were downloaded on 2022-July-06.

Proteomic labeling changes over 5 time points (0, 1, 5, 10, and 30 min) were scored per protein by fitting a cubic polynomial curve through the data using R built in functions lm and poly. A categorical term for replicate batch was included in the model, and time was continuous but transformed from minutes to ranks (1 through 5). $P$ values were calculated using F-tests comparing the models with and without the polynomial time terms. The time point with the largest deviation of mean intensity from time point 0 was used to calculate the log2FC as the difference in mean log2 intensities. Time courses of protein labeling changes were clustered by starting with a protein-by-time matrix of differences in mean log2 intensity from time 0, calculating protein–protein distances as $1-R$, where $R$ is Pearson's correlation coefficient, and defining clusters with the pam function in package cluster (version 2.1.3), with option nstart set to 20.

Networks for proteins represented in the transient and sustained cluster were generated extracting interactions between proteins in each cluster with a confidence above 0.4 derived from STRING (https://string-db.org/) (Szklarczyk et al, 2011). Interactions were important into Cytoscape (version 3.8.1) for visualization (Shannon et al, 2003). The maximum log2-fold change compared to time point 0 for each protein was visualized in the network as node color. Enrichment of Reactome pathways (https://reactome.org/) (Gillespie et al, 2022) was performed using Enrichr (https://maayanlab.cloud/Enrichr/) (Kuleshov et al, 2016; Xie et al, 2021; Chen et al, 2013).

### CRISPR knockouts

Polyclonal KO cells were generated by electroporation of Cas9 ribonucleoprotein complexes (Cas9 RNPs) into HEK293T cells stably expressing 5HT2A-APEX. Electroporation was performed using the SF Cell Line 4D-Nucleofector X Kit S (Lonza) and 4D-Nucleofector (Lonza). Recombinant S. pyogenes Cas9 protein used in this study contains two nuclear localization signal (NLS) peptides that facilitate transport across the nuclear membrane. The protein was obtained from the QB3 Macrolab, University of California, Berkeley. Purified Cas9 protein was stored in 20 mM HEPES at pH 7.5 plus 150 mM potassium chloride, 10% glycerol, and 1 mM tris(2-carboxyethyl)phosphine (TCEP) at −80 °C. Each crRNA and the tracrRNA was chemically synthesized (Dharmacon/Horizon) and suspended in 10 mM Tris-HCl pH 7.4 to generate 160 µM RNA stocks. To prepare Cas9 RNPs, crRNA, and tracrRNA were first mixed 1:1 and incubated 30 min at 37 °C to generate 80 µM crRNA:tracrRNA duplexes. An equal volume of 40 µM S. pyogenes Cas9-NLS was slowly added to the crRNA:tracrRNA and incubated for 15 min at 37 °C to generate 20 µM Cas9 RNPs. crRNAs targeting selected proteins and non-targeting controls were designed by Dharmacon. For each reaction, roughly $5 \times 10^5$ HEK293 cells were pelleted and suspended in 20 µL nucleofection buffer. 4 µL 20 µM Cas9 RNP mix was added directly to these cells and the entire volume transferred to the bottom of the reaction cuvette. Cells were electroporated using program CM-130 on the Amaxa 4D-Nucleofector (Lonza). 80 µL pre-warmed complete DMEM was added to each well and the cells were allowed to initially recover for 10 min at 37 °C followed by diluting with DMEM media, plating into 24-well plates and further recovering for 3 days. KO for selected genes was confirmed by Inference of CRISPR Edits (ICE).

## Data availability

The mass spectrometry proteomics data have been deposited to the ProteomeXchange Consortium via the PRIDE (https://www.ebi.ac.uk/pride/) (Deutsch et al, 2023) partner repository with the dataset identifier PXD040762. The source data of confocal images in Fig. 2 and Fig. 4 are submitted to BioImage Archive with

the accession number of S-BIAD1088. The source data of western blots in Fig. 2E and Fig. 6E are submitted to BioStudies with the accession number of S-BSST1404.

The source data of this paper are collected in the following database record: biostudies:S-SCDT-10_1038-S44320-024-00049-2.

## Peer review information

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

## Acknowledgements

This work was supported by funding from the Defense Advanced Research Projects Agency (DARPA) under the Cooperative Agreements HR0011-19-2-0020 (to NJK, RH, MvZ, BLR) and HR0011-20-2-0029 (to NJK, RH, BLR). The views, opinions, and/or findings contained in this material are those of the authors and should not be interpreted as representing the official views or policies of the Department of Defense or the U.S. Government. This work further received funding from the NIH (R01DA056354 to RH and MvZ; P01HL146366 to NJK; R37DA045657 to BLR). The work was carried out in the Thermo Fisher Scientific Mass Spectrometry Facility for Disease Target Discovery at the J David Gladstone Institutes and the UCSF Center for Advanced Technology. The imaging data for this study were acquired at the Center for Advanced Light Microscopy-Nikon Imaging Center at UCSF. We thank Ajda Rojc and Michael McGregor for the help with flow cytometry analysis and members of Krogan lab for helpful advice and comments.

## Author contributions

**Xiaofang Zhong**: Conceptualization; Data curation; Formal analysis; Validation; Investigation; Visualization; Writing—original draft; Writing—review and editing. **Qiongyu Li**: Conceptualization; Data curation; Formal analysis; Investigation; Visualization; Writing—original draft; Writing—review and editing. **Benjamin J Polacco**: Data curation; Formal analysis; Investigation; Visualization; Methodology; Writing—review and editing. **Trupti Patil**: Formal analysis; Investigation; Visualization; Methodology. **Aaron Marley**: Validation; Investigation. **Helene Foussard**: Formal analysis; Investigation; Visualization. **Prachi Khare**: Formal analysis; Investigation. **Rasika Vartak**: Formal analysis; Investigation; Visualization. **Jiewei Xu**: Formal analysis; Validation; Investigation; Methodology. **Jeffrey F DiBerto**: Formal analysis; Investigation; Visualization. **Bryan L Roth**: Supervision; Funding acquisition; Writing—review and editing. **Manon Eckhardt**: Data curation; Writing—original draft. **Mark von Zastrow**: Conceptualization; Supervision; Funding acquisition; Writing—original draft; Writing—review and editing. **Nevan J Krogan**: Supervision; Funding acquisition. **Ruth Hüttenhain**: Conceptualization; Data curation; Formal analysis; Supervision; Funding acquisition; Validation; Investigation; Visualization; Methodology; Writing—original draft; Project administration; Writing—review and editing.

Source data underlying figure panels in this paper may have individual authorship assigned. Where available, figure panel/source data authorship is listed in the following database record: biostudies:S-SCDT-10_1038-S44320-024-00049-2.

## Disclosure and competing interests statement

The Krogan Laboratory has received research support from Vir Biotechnology, F Hoffmann-La Roche, and Rezo Therapeutics. NJK has a financially compensated consulting agreement with Maze Therapeutics. NJK is the President and is on the Board of Directors of Rezo Therapeutics, and he is a shareholder in Tenaya Therapeutics, Maze Therapeutics, Rezo Therapeutics, and Interline Therapeutics. BLR is a member of the Scientific Advisory Boards of Septerna Pharmaceuticals, Escient Pharmaceuticals, Onsero Pharmaceuticals, and Levator.

