## [Peer Review File · Molecular Systems Biology]

A proximity proteomics pipeline with improved reproducibility and throughput

Xiaofang Zhong, Qiongyu Li, Benjamin Polacco, Trupti Patil, Aaron Marley, Helene Foussard, Prachi Khare, Rasika Vartak, Jiewei Xu, Jeffrey DiBerto, Bryan Roth, Manon Eckhardt, Mark von Zastrow, Nevan Krogan, and Ruth Huttenhain

Corresponding author(s): Ruth Huttenhain (ruthh@stanford.edu)

Review Timeline:

Submission Date:	7th Apr 23
Editorial Decision:	17th May 23
Revision Received:	26th Apr 24
Editorial Decision:	5th Jun 24
Revision Received:	9th Jun 24
Accepted:	11th Jun 24

Editors: Maria Polychronidou and Jingyi Hou

Transaction Report:

17th May 2023

Manuscript Number: MSB-2023-11697

Title: An automated proximity proteomics pipeline for subcellular proteome and protein interaction mapping

Dear Ruth,

Thank you again for submitting your work to Molecular Systems Biology. We have now heard back from the four referees who agreed to evaluate your study. As you will see below, the reviewers raise a series of concerns, which preclude the publication of your study in its current form. Overall, the reviewers think that as it stands the study seems somewhat preliminary.

However, given that the reviewers acknowledge that the presented pipeline and findings seem relevant, we have decided to offer you the chance to address the issues raised in a major revision. Without repeating all the points raised by the reviewers, some of the more fundamental issues are the following:

- The reviewers point out that as it stands the pipeline is semi-automated. Fully automating the process would significantly enhance the impact of the study.
- The level of biological insight remains rather limited. Further analyses providing new insights into the serotonin receptor network need to be included.
- The reviewers also list several technical concerns which need to be convincingly addressed.
- In line with the comment of reviewer #2 and given the focus of the study on the presented pipeline, we would ask you to make sure that sufficient methodological details are included so that future users can easily adopt the pipeline. Step-by-step protocols can be included in the Materials and Methods (please see also the point below regarding Structured Methods) or in the Appendix or protocols.io if they are very lengthy.

All issues raised by the reviewers need to be satisfactorily addressed. The reviewers make constructive suggestions on how to improve the study. As you may already know, our editorial policy allows in principle a single round of major revision. It is therefore essential to provide responses to the reviewers' comments that are as complete as possible. If you have any questions or if you would like to discuss your revision plan with me please feel free to get in touch.

On a more editorial level, we would ask you to address the following points:

- Please include 5 keywords.
- Please provide a .doc file for the manuscript text (including legends for the main figures) and individual production-quality files for the main figures (one file per figure).
- We have replaced Supplementary Information by the Expanded View (EV format). In this case, all additional figures can be included in a PDF called Appendix. Appendix Figures should be labeled and called out as: "Appendix Figure S1, Appendix Figure S2..." etc. Each legend should be below the corresponding Figure/Table in the Appendix. Please include a Table of Contents in the beginning of the Appendix. For detailed instructions regarding expanded view please refer to our Author Guidelines: .
- Tables S1-S5 should be provided as EV Tables (short tables, < 1 page long) or EV Datasets (long or complex tables). Please provide one file per EV Table/Dataset and include in each .xls file a description of the Table/Dataset in a separate tab.
- Please provide a "standfirst text" summarizing the study in one or two sentences (approximately 250 characters), three to four "bullet points" highlighting the main findings and a "synopsis image" (550px width and max 400px height, jpeg format) to highlight the paper on our homepage.
- All Materials and Methods need to be described in the main text. We would ask you to use 'Structured Methods', our new Materials and Methods format, which is mandatory for Methods or Articles with a strong methodological focus. According to this format, the Materials and Methods section should include a Reagents and Tools Table (listing key reagents, experimental models, software and relevant equipment and including their sources and relevant identifiers) followed by a Methods and Protocols section in which we encourage the authors to describe their methods using a step-by-step protocol format with bullet points, to facilitate the adoption of the methodologies across labs. More information on how to adhere to this format as well as downloadable templates (.doc or .xls) for the Reagents and Tools Table can be found in our author guidelines: . An example of a Method paper with Structured Methods can be found here:

- Please include a Data availability section describing how the data have been made available. This section needs to be formatted according to the example below:
The datasets and computer code produced in this study are available in the following databases:
 - Chip-Seq data: Gene Expression Omnibus GSE46748 (<https://www.ncbi.nlm.nih.gov/geo/query/acc.cgi?acc=GSE46748>)
 - Modeling computer scripts: GitHub (<https://github.com/SysBioChalmers/GECKO/releases/tag/v1.0>)
 - [data type]: [full name of the resource] [accession number/identifier] ([doi or URL or identifiers.org/DATABASE:ACCESSION])
 - Please include a "Disclosure & Competing Interests Statement" in the man text.
 - For data quantification: please specify the name of the statistical test used to generate error bars and P values, the number (n) of independent experiments (specify technical or biological replicates) underlying each data point and the test used to calculate p-values in each figure legend. The figure legends should contain a basic description of n, P and the test applied. Graphs must include a description of the bars and the error bars (s.d., s.e.m.).
 - When you resubmit your manuscript, please download our CHECKLIST (<https://bit.ly/EMBOPressAuthorChecklist>) and include the completed form in your submission.
- *Please note* that the Author Checklist will be published alongside the paper as part of the transparent process (<https://www.embopress.org/page/journal/17444292/authorguide#transparentprocess>).

If you feel you can satisfactorily deal with these points and those listed by the referees, you may wish to submit a revised version of your manuscript. Please attach a covering letter giving details of the way in which you have handled each of the points raised by the referees. A revised manuscript will be once again subject to review and you probably understand that we can give you no guarantee at this stage that the eventual outcome will be favorable.

Kind regards,

Maria

Maria Polychronidou, PhD
Senior Editor
Molecular Systems Biology

We realize that it is difficult to revise to a specific deadline. In the interest of protecting the conceptual advance provided by the work, we recommend a revision within 3 months (15th Aug 2023). Please discuss the revision progress ahead of this time with the editor if you require more time to complete the revisions. Use the link below to submit your revision:

IMPORTANT: When you send your revision, we will require the following items:

1. the manuscript text in LaTeX, RTF or MS Word format
 2. a letter with a detailed description of the changes made in response to the referees. Please specify clearly the exact places in the text (pages and paragraphs) where each change has been made in response to each specific comment given
 3. three to four 'bullet points' highlighting the main findings of your study
 4. a short 'blurb' text summarizing in two sentences the study (max. 250 characters)
 5. a 'thumbnail image' (550px width and max 400px height, Illustrator, PowerPoint or jpeg format), which can be used as 'visual title' for the synopsis section of your paper.
 6. Please include an author contributions statement after the Acknowledgements section (see <https://www.embopress.org/page/journal/17444292/authorguide>)
 7. Please complete the CHECKLIST available at (<https://bit.ly/EMBOPressAuthorChecklist>). Please note that the Author Checklist will be published alongside the paper as part of the transparent process (<https://www.embopress.org/page/journal/17444292/authorguide#transparentprocess>).
 8. When assembling figures, please refer to our figure preparation guideline in order to ensure proper formatting and readability in print as well as on screen:
<https://bit.ly/EMBOPressFigurePreparationGuideline>
See also figure legend guidelines: <https://www.embopress.org/page/journal/17444292/authorguide#figureformat>
9. Please note that corresponding authors are required to supply an ORCID ID for their name upon submission of a revised manuscript (EMBO Press signed a joint statement to encourage ORCID adoption). (<https://www.embopress.org/page/journal/17444292/authorguide#editorialprocess>)
Currently, our records indicate that the ORCID for your account is 0000-0002-0896-5910.

Link Not Available

The system will prompt you to fill in your funding and payment information. This will allow Wiley to send you a quote for the article processing charge (APC) in case of acceptance. This quote takes into account any reduction or fee waivers that you may be eligible for. Authors do not need to pay any fees before their manuscript is accepted and transferred to the publisher.

EMBO Press participates in many Publish and Read agreements that allow authors to publish Open Access with reduced/no publication charges. Check your eligibility: <https://authorservices.wiley.com/author-resources/Journal-Authors/open-access/affiliation-policies-payments/index.html>

*** PLEASE NOTE *** As part of the EMBO Press transparent editorial process initiative (see our Editorial at <https://dx.doi.org/10.1038/msb.2010.72>), Molecular Systems Biology publishes online a Review Process File with each accepted manuscripts. This file will be published in conjunction with your paper and will include the anonymous referee reports, your point-by-point response and all pertinent correspondence relating to the manuscript. If you do NOT want this File to be published, please inform the editorial office at msb@embo.org within 14 days upon receipt of the present letter.

Reviewer #1:

This manuscript described the development of a scalable proximity labeling (PL) proteomics pipeline and demonstrated its applicability in large scale PL experiments capturing proteins from various subcellular compartments and dynamics of serotonin trafficking in HEK cells. This study automated the biotinylation enrichment and washing steps using a KingFisher Flex system with improved precision, throughput, and protein IDs compared to manual processing. Sample preparation and DIA-MS parameters were carefully optimized for improved protein coverage and reproducibility. Using this integrated pipeline, the authors comprehensively mapped subcellular proteomes using 5 PL-probes localized to the plasma membrane, Golgi, endosome, and late endosome/lysosome. Furthermore, the temporal dynamics of serotonin receptor interactions and trafficking were characterized in a ligand and time-dependent manner, illustrating both transient and sustained changes.

Overall, this is a very elegant study with solid method development and comprehensive applications. The experiments and PL data filtering criteria are well-designed, generating reliable conclusions. Proximity labeling proteomics typically involves intensive manual sample processing with limited scalability using a magnetic rack. This automated process is universally applicable to proximity labeling proteomics and can be a very useful resource for the field. The use of DIA-MS further reduced missing values, which is an important improvement for large scale PL-proteomics.

However, I do have some concerns that can be addressed during revision. Given the popularity of liquid handler equipment and existing fully automated sample preparation platforms, this study only automated the biotinylation enrichment and washing steps. Although very useful, authors should recognize that this is a semi-automated pipeline. Another concern is about the biological insights for serotonin receptor interactions. How can these identified interactions be used to provide new insights about serotonin networking is unclear. Using HEK cell line also limited the discovery since serotonin network is most relevant in neurons, which has been recognized by the authors in the discussion section.

Major comments

1. As mentioned above, only the biotinylation enrichment and washing steps are automated. Since proteins are still bound on the magnetic beads after washing, why not automate the entire process including reduction, alkylation, and even on-beads digestion and peptide cleanup? Recently published automation manuscripts typically provided fully automated sample preparation pipeline (for example recent publications from Kringsveld and Van Eyk groups).
2. The decreased plasma membrane proteins and increased endosomal proteins during temporal serotonin trafficking as well as the transient and sustained interactions are very exciting findings that really only possible using PL-proteomics. I am curious about how these data can be used to further advance the biological understanding of serotonin trafficking and its relevant to biomedical research since serotonin can be used to treat depression and some muscle disorders. Can the authors provide some discussions on this aspect?
3. From the serotonin tracking dataset, are there any key proteins that are worth following up or validate? Transient interactions are indeed difficult to validate, but authors can provide some discussions or future remarks.
4. Streptavidin is a major abundant contaminant in PL-proteomics data. Can the author provide the identified m/z of streptavidin peptides and compare the number of streptavidin peptides/precursors in manual vs. automated data, and in DDA vs. DIA data in supplemental figures? Since DIA fragments all masses within the wide isolation window, I am concerned that with DIA, abundant

streptavidin will appear in most MS/MS scans hindering the confident identification and accurate quantification of other peptides.

5. The authors carefully optimized a series of DIA parameters. Can the author also provide the rationale or optimization results that leads to the selection of 20 Da isolation window which seems large. Since PL-proteomics data is less complex compared to whole cell lysate, a smaller isolation window may benefit the accuracy of DIA identification and quantification.

6. In the comparison of autoDIA vs. manualDIA, how do protein intensities correlate between autoDIA vs. manualDIA? Although we cannot compare intensities between DDA vs. DIA, it would be helpful to show how protein quantification ratios (for example APEX vs. control) correlate between DIA vs. DDA?

Minor comments:

1. It is a valid argument to select 80 uL instead of 100 uL of beads per 1 mg of proteins to reduce nonspecific binding. However, the way to evaluate nonspecific bindings needs to be revised. More beads will of course enrich more proteins in the negative control group, also more proteins in the PM-APEX probe. The authors can provide the volcano plots or Venn diagram of PM-APEX vs control for different beads amount and highlight some nonspecific proteins in the plots. Or comparing PM-APEX 80 uL vs. 100 uL beads to show that key proteins are still enriched but maybe with more nonspecific bindings in 100 uL.
2. Can the authors label the bait proteins and maybe also several key proteins in the volcano plots in Figure 4C and provide the number of proteins used in Figure 4F for each subcellular compartment? Can the author also label the number of transient and sustained proteins in Figure 5D?
3. Figure S2 needs to be adjusted to reveal the subtle differences of different comparison groups. For example, the authors can start the y axis with 10K or 20K peptides instead of 0. Can the author also provide the peptide IDs in Figure S2A and B?

Reviewer #2:

In this paper, Zhong and colleagues present a high-throughput and reproducible pipeline for proximity labeling proteomics. They employ several quality-control steps during generation of clones with an APEX2-tagged protein of interest, including using flow cytometry to determine expression consistency, immunofluorescence to check tagged protein localization, and Western blot analysis of biotinylation efficiency. These cell line validation steps are relatively standard in the field and would be more rigorous if they included checking the expression level of the APEX2-tagged protein relative to the endogenous level. The authors next optimize and validate a partially automated pipeline for proximity labeling using the KingFisher Flex to perform biotinylated protein isolation and on-bead digestion and mass spectrometry sample preparation, followed by a comparison of data-independent acquisition (DIA) and data-dependent acquisition (DDA) mass spectrometry analysis. DIA in their automated format allowed more precursor ion identifications, had lower coefficients of variation, and was more reproducible, based on three replicates.

To test their pipeline's performance in subcellular proteome mapping, the authors generate separate cell lines with Lyn11 (plasma membrane), 2xFYVE (endosome), LAMTOR/LAMP1 (lysosome), GalT (Golgi), and a nuclear export signal (control) tagged with APEX2. They find many expected proteins based on GO enrichment analysis and comparison to the benchmark Human Cell Map (Go et al. 2021). Several proteins were identified in the new data not found in the Human Cell Map reference, including Golgin-45 in the Golgi, as expected based on other lines of published evidence. As a further proof-of-principle experiment-this time tracking the serotonin 2A receptor-Zhong and colleagues generated HEK293T cells expressing 5HT2A-APEX2 and performed a proximity labeling time course after addition of the 5HT ligand. They find a decrease in plasma membrane protein labeling over the time course and an increase in endosomal and lysosomal protein labeling over the time course, suggesting movement of the receptor through the endocytic system. This is analogous to previous work (2017 if I remember correctly) demonstrating the use of proximity biotinylation to follow trafficking of membrane proteins (e.g. GPCRs) through the endolysosomal system, including a paper by the authors.

Major points:

- The authors do not compare expression levels of APEX2-tagged proteins of interest with endogenous proteins. This is known to be an important factor in the specificity of proximity proteomics and can significantly affect the results. The authors themselves state, "both the optimal protein lysate starting amount and streptavidin bead amount might be dependent on the expression level of the PL construct and the cell type used," but they do not show a figure comparing expression levels. Since the majority of time would be spent optimizing cell lines for expression levels, it isn't clear that the methods are as effective as advertised in speeding up analyses, although reproducibility could be enhanced with the method.
- The subcellular proximity proteomics experiments and serotonin receptor experiments do not appear to find any novel protein localizations, and no follow-up experiments are presented. Furthermore, these experiments are relatively small in size, and the 96-well format using the KingFisher does not seem strictly necessary. Because the Human Cell Map was published in 2021 and used 192 baits, this paper does not appear to present a major advance.
- The initial cell line generation and validation section of the paper implies that this process is somehow an advance on existing protocols or is faster in some ways than previous methods. However, the methods as described in the manuscript are relatively standard and do not seem to employ any automation that would speed up the process. Automation is not necessary, but the manuscript should be revised so as not to give the impression that this represents a new and improved cell line generation pipeline.

- Numerous publications have already compared DDA- and DIA-based mass spectrometry analysis. The authors should include further discussion of how applying this to proximity proteomics presents an advance.

Minor points:

- At several points in the manuscript, references for known protein localizations and 5HT2A interactors appear to be missing (see the last few sentences of subheading "Mapping subcellular proteomes using the automated proximity proteomics strategy" and the last paragraph of subheading "Mapping ligand-dependent proximal interaction network changes for the 5HT2A receptor").
- The authors mention that their pipeline "could be further optimized by shortening the time for binding of the biotinylated proteins to streptavidin beads". Including data for shorter binding times and comparing it to the current overnight binding time would increase the efficiency and usefulness of this pipeline.
- In the discussion, the authors point out that the 5HT2A receptor is not expressed in HEK293T cells used here and that another, more relevant cell type would be more informative. Perhaps including this type of experiment would strengthen the paper if new validated interactions could be identified.
- APEX2-tagged protein localization shown in Figure 4A shows non-specific localization, especially of 2xFYVE. The authors should discuss why this is the case and whether localization can be improved. It is also unclear whether binding of overexpressed FYVE domains that bind to phospholipids would alter the kinetics of endosomal maturation/signaling.
- The manuscript is described as a step-by-step guide for proximity proteomics. However, several areas lack enough detail to be a true guide.
- The data in Figure 3 presumably only shows data at the protein level, not the precursor or peptide level. This should be stated explicitly somewhere, and it may be helpful to see precursor level data to support the author's statement that Auto_DIA outperformed the other methods in terms of precursor identification.
- There seem to be only two replicates of the 2xFYVE experiment. Would the authors please explain why there are not three replicates?
- The GO analysis in the right part of Figure 4E is a bit confusing since it uses the same blue color as the heat map in the left of Figure 4E. Perhaps using the color scheme in Figure 5B for Figure 4E would be more clear.
- The time point used for the analysis in Figure 5C should be specified in the text and/or figure legend.
- In the discussion, the authors state the "DIA-based protein quantification already improved the reproducibility in protein identification from 78.7% to 95% overlap among three replicates...". However, it appears that reproducibility was not assessed at the protein level, so this statement is slightly misleading. Would the authors please clarify?

Comments to the editor

Overall, the pipeline presented in this manuscript presents a modest advance in proximity proteomics. This is a relatively high throughput sample processing method coupled with DIA-based mass spectrometry, but it still relies on labor-intensive cell line generation prior to proximity labeling. Furthermore, the applications presented here appear to validate the pipeline but do not uncover new information about subcellular proteomes or about serotonin signaling. In my view, a true advance would be to use the pipeline to survey hundreds of baits in order to rival the Cell Map experiment performed in the Gingras lab several years ago.

Reviewer #3:

In this study, Zhong and Li et al. developed and benchmarked an automated pipeline, as well as cell line reagents, for APEX2-based spatially-resolved proteomics in cultured cells. This pipeline increased sample processing throughput and also reduced variability. The authors demonstrated its utility by profiling proteomes of several cellular compartments (inner surface of the plasma membrane, endo/lysosome, and Golgi) and temporally-resolved interactomes of the serotonin receptor 5HT2A upon agonist activation. This pipeline and dataset would be of interest to a broad community of cell biologists. Below please find specific points:

- The authors wrote that "...while clone 1 is mainly diffused in the cytoplasm (Figure 2D), an observation also recapitulated in the wider fluorescence distribution seen by flow cytometry for this clone." A wider fluorescence intensity distribution on FACS has nothing to do with the cytosolic diffusion of APEX2 in Clone 1. It only means that the cell population is more heterogeneous with variable expression levels. The authors should not mistakenly connect them.
- The authors wrote that "While 80 μ L of streptavidin beads resulted in slightly lower protein identifications compared to 100 μ L, the number of identified proteins in the negative control (without H₂O₂ treatment) was reduced using 80 μ L of beads, indicating less unspecific binding to beads (Figure S2B)." Based on Figure S2B, the difference between -H₂O₂ 80uL (2nd gray bar) and -H₂O₂ 100uL (3rd gray bar) was very small while the difference between +H₂O₂ 80uL (2nd blue bar) and +H₂O₂ 100uL (3rd blue bar) seems larger. Practically, 80uL and 100uL seem no difference overall. The authors should remove the 80/100uL comparison here.
- In Figure 4A, the 2xFYVE GFP staining has a much broader expression compared with Rab5, indicating non-endosome

localization. The authors should comment on this and the potential caveats caused by it. The GalT (Golgi) one has a similar problem but is likely less severe.

- As a resource/method paper, the authors should consider depositing the constructs and cell lines made in this study to Addgene or other repositories.

Reviewer #4:

Review notes for: An automated proximity proteomics pipeline for subcellular proteome and protein interaction mapping in Molecular Systems Biology

Summary:

This paper presents a workflow for the preparation and analysis of APEX2-based proximity interaction proteomics. The aim is to construct a framework for other researchers to quickly perform proximity-dependent experiments. The core of the workflow is a method using a KingFisher liquid handler to do biotinylated protein enrichment. This is followed by DIA-MS analysis as it was shown to be superior to traditional DDA analysis. The workflow is demonstrated by confirming previous Cell Map localizations and then by constructing a time-course of the response of a serotonin receptor after activation by 5-hydroxytryptamine.

General Remarks:

The study's outcomes align with prior research and make sense, with the authors providing plausible interpretations. The paper showcases a technical progression and could serve as a valuable resource for researchers aiming to enhance the throughput of their proximity-labelling datasets. The biological findings, however, are not as prominent as the technical accomplishments.

In conclusion, while this manuscript is acceptable in principle (once the issues below are addressed), it is in a nascent stage. It leaves much to be desired due to the unexplored potential within the premise of an "automated method". Indeed, the workflow's automation is minimal, predominantly focusing on a single portion of the experiment. Several upstream molecular biology steps, crucial for obtaining good results, are left manual and could greatly benefit from automation. The title, promising an automated proximity biotinylation workflow, is somewhat misleading as it implies more comprehensive automation than what is presented in the study.

Major points:

Numerous times throughout the manuscript, including the title, refer to the presented pipeline as 'The manuscript repeatedly designates the proposed pipeline as 'automated' in several instances, including the title. However, this is somewhat misleading as only a fraction of the pipeline is automated. The bulk of the procedure, including steps like cell lysis (and possibly digestion-it's unclear), which could feasibly be conducted on the liquid handler, remains manual. Consideration of the cell line generation in an automated or semi-automated manner is not presented.

Furthermore, the power of an automated platform for method optimization is almost completely neglected. For example, only a single protein load was tested against a handful of streptavidin particle amounts. There was a desire to reduce the background intensity of non-specific captured proteins, yet only a single very long initial binding time (12 hours) was explored. This is disappointing. An automated platform allows a broad range of different parameters to be tested with ease, but in this case, only very limited ones are attempted. It might be possible that the authors conducted a meticulous optimization of parameter selection before their work, but this isn't presented in the manuscript. If this is the case, I suggest including this data to assist other researchers in identifying key parameters.

The choice and presentation of the mass spectrometry methods are less than ideal. For instance, during their method optimization, they analyze DIA data in a library-free fashion. However, for their actual biological experiments, they depend on spectral library-based analysis. The reasoning behind this decision is unclear, as the results from a library-free analysis can significantly differ from those where a spectral library limits the types of peptides discovered.

The absence of statistical comparisons in their initial method optimization results is puzzling, especially given the effective application of statistics to their proximity labelling data. They have three methods with three replicates but only report the mean (presumably-it's not specified) for the various numbers of identifications. Why not include the coefficient of variation (CV), and potentially compare these three methods statistically using a basic tool like the Students' t-test?

Minor point:

The language surrounding the MS acquisition and analysis parameters could be made clearer. For example, 1) the samples were dried by vacuum centrifugation, but they must have been resuspended in something to be injected on the LC. What was used for this resuspension? How much of the sample was injected? Was the injection performed directly from the well plate? 2) Was the digestion process carried out using the liquid handler? I suggest that the authors carefully revisit the presentation of their methods.

Suggestions/Comments: shown in black.

Responses: shown in blue.

Point-by-point response to Editorial Office

Thank you again for submitting your work to Molecular Systems Biology. We have now heard back from the four referees who agreed to evaluate your study. As you will see below, the reviewers raise a series of concerns, which preclude the publication of your study in its current form. Overall, the reviewers think that as it stands the study seems somewhat preliminary.

However, given that the reviewers acknowledge that the presented pipeline and findings seem relevant, we have decided to offer you the chance to address the issues raised in a major revision. Without repeating all the points raised by the reviewers, some of the more fundamental issues are the following:

- The reviewers point out that as it stands the pipeline is semi-automated. Fully automating the process would significantly enhance the impact of the study.

We agree with the reviewer that it would be beneficial to fully automate the protocol including the enzymatic digestion and C18 clean up. There are a few studies that have described automated digestion and clean up of samples, for example:

- <https://www.ncbi.nlm.nih.gov/pmc/articles/PMC6845026/>
- <https://www.embopress.org/doi/full/10.15252/msb.20199111>
- <https://www.ncbi.nlm.nih.gov/pmc/articles/PMC8210952/#S10>

Unfortunately, these fully automated workflows require additional equipment, such as a liquid handler, which we do not have available. However, to address the reviewers comments we (1) clarified that the protocol in its current version is semi-automated, but given the performance in 96-well plate the throughput and reproducibility of the digestion and C18 clean up is still improved and (2) we have added information and references for additional automation of the protocol.

- The level of biological insight remains rather limited. Further analyses providing new insights into the serotonin receptor network need to be included.

As our paper is a method, the overall focus of the current study is on the development of a semi-automated, higher throughput strategy for proximity labeling and demonstrating its application to different biological questions. As a proof of concept, we successfully apply the pipeline to common PL applications: mapping compartment specific proximal proteomes and mapping dynamics of protein interaction networks, based on the example of the serotonin 2A receptor (5HT2A) interaction network. To address the comments of the reviewers related to new biological insights on the serotonin receptor network, we decided that the best fit for this method paper would be to perform on a proof of concept how we can utilize our PL pipeline to start interrogating selected 5HT2A network components. While we demonstrate this for the 5HT2A receptor, the strategy is broadly applicable to other targets.

Specifically, our goal was to examine how CRISPR-based knockout (KO) of individual network components affect the overall activity-dependent 5HT2A interaction network. To accomplish this, we first demonstrate how the semi-automated PL strategy can be leveraged to perform PL experiments at a smaller scale, 6-well plate format compared to 10cm dish. This is a significant

advance as it overcomes another bottleneck for large-scales PL studies, the cell culture scale. We then optimize a combined CRISPR knockout-PL strategy to knock out two selected 5HT_{2A} interactors, ANXA2 and S100A10, which have been previously described to form heterotetramers that bind to membrane proteins, assisting their trafficking to the plasma membrane. Our data revealed that the knockout of both, Annexin A2 (ANXA2) and S100-A10 (S100A10) resulted in reduced biotinylation of ANXA2, S100A10, and AHNAK, which is consistent with previous findings demonstrating that these proteins physically interact. Finally, we also investigate the effect of ANXA2 and S100A10 KO on cell surface expression of the 5HT_{2A} receptor, which revealed a small but significant decrease in the expression of the 5HT_{2A} receptor at the plasma membrane in both ANXA2 and S100A10 KO cells.

The new data is presented in a completely new section at the end of the manuscript, with an associated new main and extended data figure. We believe that this section represents a significant addition strengthening our method paper as it demonstrates the efficacy of the automated, low-input PL protocol (1) in delivering reproducible results at a smaller scale and (2) in monitoring interaction network perturbations across multiple conditions with CRISPR KO of proximal interactors.

We intentionally do not include more information about how activity-dependent interactions of 5HT_{2A} might affect receptor function as these would be more speculations at this point which would require functional follow up experiments. While these would indeed be interesting to perform, and we agree that it would be important to include functional studies in neuronal systems, we believe that these are beyond the scope of our manuscript, which is focused on the method and its performance.

- The reviewers also list several technical concerns which need to be convincingly addressed. We believe that we have addressed all technical concerns that were brought up by the reviewers in the revised version. See details in the responses to the reviewer's concerns below.

- In line with the comment of reviewer #2 and given the focus of the study on the presented pipeline, we would ask you to make sure that sufficient methodological details are included so that future users can easily adopt the pipeline. Step-by-step protocols can be included in the Materials and Methods (please see also the point below regarding Structured Methods) or in the Appendix or protocols.io if they are very lengthy.

We thank the reviewer for the suggestion and agree that it would be very useful to provide a step by step protocol. We have addressed this comment by (1) adding additional information to the method section to clarify experimental and computational workflows and (2) putting together a detailed protocol on protocols.io: [dx.doi.org/10.17504/protocols.io.yxmvm3jbb13p/v1](https://doi.org/10.17504/protocols.io.yxmvm3jbb13p/v1)

All issues raised by the reviewers need to be satisfactorily addressed. The reviewers make constructive suggestions on how to improve the study. As you may already know, our editorial policy allows in principle a single round of major revision. It is therefore essential to provide responses to the reviewers' comments that are as complete as possible. If you have any questions or if you would like to discuss your revision plan with me please feel free to get in touch.

- Please include 5 keywords.

Proximity proteomics, APEX-based proximity labeling, protein-protein interaction, subcellular proteomics, G protein-coupled receptor

- Please provide a .doc file for the manuscript text (including legends for the main figures) and individual production-quality files for the main figures (one file per figure).

We provide a doc file for the manuscript text and high quality figures for the main figures.

- We have replaced Supplementary Information with the Expanded View (EV format). In this case, all additional figures can be included in a PDF called Appendix. Appendix Figures should be labeled and called out as: "Appendix Figure S1, Appendix Figure S2..." etc. Each legend should be below the corresponding Figure/Table in the Appendix. Please include a Table of Contents in the beginning of the Appendix. For detailed instructions regarding expanded view please refer to our Author Guidelines: <<http://msb.embopress.org/authorguide#expandedview>>.

- Tables S1-S5 should be provided as EV Tables (short tables, < 1 page long) or EV Datasets (long or complex tables). Please provide one file per EV Table/Dataset and include in each .xls file a description of the Table/Dataset in a separate tab.

- Please provide a "standfirst text" summarizing the study in one or two sentences (approximately 250 characters), three to four "bullet points" highlighting the main findings and a "synopsis image" (550px width and max 400px height, jpeg format) to highlight the paper on our homepage.

We have provided all these items.

- All Materials and Methods need to be described in the main text. We would ask you to use 'Structured Methods', our new Materials and Methods format, which is mandatory for Methods or Articles with a strong methodological focus. According to this format, the Materials and Methods section should include a Reagents and Tools Table (listing key reagents, experimental models, software and relevant equipment and including their sources and relevant identifiers) followed by a Methods and Protocols section in which we encourage the authors to describe their methods using a step-by-step protocol format with bullet points, to facilitate the adoption of the methodologies across labs. More information on how to adhere to this format as well as downloadable templates (.doc or .xls) for the Reagents and Tools Table can be found in our author guidelines:

<<https://www.embopress.org/page/journal/17444292/authorguide#textformat>>. An example of a Method paper with Structured Methods can be found here: <<https://www.embopress.org/doi/10.15252/msb.20178071>>.

The Material & Method section contains a Reagents and Tools section. We also provide a step by step protocol for the proximity labeling protocol on protocols.io.

- Please include a Data availability section describing how the data have been made available. This section needs to be formatted according to the example below:

The datasets and computer code produced in this study are available in the following databases:

- Chip-Seq data: Gene Expression Omnibus GSE46748
(<https://www.ncbi.nlm.nih.gov/geo/query/acc.cgi?acc=GSE46748>)

- Modeling computer scripts: GitHub
(<https://github.com/SysBioChalmers/GECKO/releases/tag/v1.0>)

- [data type]: [full name of the resource] [accession number/identifier] ([doi or URL or
The Data availability section has been formatted accordingly.

- Please include a "Disclosure & Competing Interests Statement" in the main text.

We have included a Disclosure & Competing Interests Statement in the main text.

- For data quantification: please specify the name of the statistical test used to generate error bars and P values, the number (n) of independent experiments (specify technical or biological replicates) underlying each data point and the test used to calculate p-values in each figure legend. The figure legends should contain a basic description of n, P and the test applied. Graphs must include a description of the bars and the error bars (s.d., s.e.m.).

We have included this information in the figure legends.

- When you resubmit your manuscript, please download our CHECKLIST
(<https://bit.ly/EMBOPressAuthorChecklist>) and include the completed form in your submission.

Please note that the Author Checklist will be published alongside the paper as part of the transparent process

(<https://www.embopress.org/page/journal/17444292/authorguide#transparentprocess>).

We have completed the author checklist.

IMPORTANT: When you send your revision, we will require the following items:

1. the manuscript text in LaTeX, RTF or MS Word format
2. a letter with a detailed description of the changes made in response to the referees. Please specify clearly the exact places in the text (pages and paragraphs) where each change has been made in response to each specific comment given
3. three to four 'bullet points' highlighting the main findings of your study
4. a short 'blurb' text summarizing in two sentences the study (max. 250 characters)
5. a 'thumbnail image' (550px width and max 400px height, Illustrator, PowerPoint or jpeg format), which can be used as 'visual title' for the synopsis section of your paper.
6. Please include an author contributions statement after the Acknowledgements section (see <https://www.embopress.org/page/journal/17444292/authorguide>)

An author contribution statement is included in the main text after the Acknowledgements section.

7. Please complete the CHECKLIST available at (<https://bit.ly/EMBOPressAuthorChecklist>). Please note that the Author Checklist will be published alongside the paper as part of the transparent process

(<https://www.embopress.org/page/journal/17444292/authorguide#transparentprocess>).

We have completed the author checklist.

See also figure legend guidelines:

<https://www.embopress.org/page/journal/17444292/authorguide#figureformat>

We have reviewed the guidelines.

9. Please note that corresponding authors are required to supply an ORCID ID for their name upon submission of a revised manuscript (EMBO Press signed a joint statement to encourage ORCID adoption).

(<https://www.embopress.org/page/journal/17444292/authorguide#editorialprocess>)

An ORCID ID will be supplied.

Source data has been gathered and uploaded.

We realize that it is difficult to revise to a specific deadline. In the interest of protecting the conceptual advance provided by the work, we recommend a revision within 3 months (15th Aug 2023). Please discuss the revision progress ahead of this time with the editor if you require more time to complete the revisions.

Point-by-point response to Reviewer #1

This manuscript described the development of a scalable proximity labeling (PL) proteomics pipeline and demonstrated its applicability in large scale PL experiments capturing proteins from various subcellular compartments and dynamics of serotonin trafficking in HEK cells. This study automated the biotinylation enrichment and washing steps using a KingFisher Flex system with improved precision, throughput, and protein IDs compared to manual processing. Sample preparation and DIA-MS parameters were carefully optimized for improved protein coverage and reproducibility. Using this integrated pipeline, the authors comprehensively mapped subcellular proteomes using 5 PL-probes localized to the plasma membrane, Golgi, endosome, and late endosome/lysosome. Furthermore, the temporal dynamics of serotonin receptor interactions and trafficking were characterized in a ligand and time-dependent manner, illustrating both transient and sustained changes.

Overall, this is a very elegant study with solid method development and comprehensive applications. The experiments and PL data filtering criteria are well-designed, generating reliable conclusions. Proximity labeling proteomics typically involves intensive manual sample processing with limited scalability using a magnetic rack. This automated process is universally applicable to proximity labeling proteomics and can be a very useful resource for the field. The use of DIA-MS further reduced missing values, which is an important improvement for large scale PL-proteomics.

We thank the reviewer for the positive evaluation of our study.

However, I do have some concerns that can be addressed during revision. Given the popularity of liquid handler equipment and existing fully automated sample preparation platforms, this study only automated the biotinylation enrichment and washing steps. Although very useful, authors should recognize that this is a semi-automated pipeline. Another concern is about the biological insights for serotonin receptor interactions. How can these identified interactions be used to provide new insights about serotonin networking is unclear. Using HEK cell line also limited the discovery since the serotonin network is most relevant in neurons, which has been recognized by the authors in the discussion section.

Major comments

1. As mentioned above, only the biotinylation enrichment and washing steps are automated. Since proteins are still bond on the magnetic beads after washing, why not automate the entire process including reduction, alkylation, and even on-beads digestion and peptide cleanup? Recently published automation manuscripts typically provided a fully automated sample preparation pipeline (for example recent publications from Kringsveld and Van Eyk groups).

As mentioned above, we agree with the reviewer that it would be beneficial to fully automate the protocol including the enzymatic digestion and C18 clean up. However, further automation as it is described in previous publications for example from the Kringsveld and Van Eyk groups would require a liquid handler. While we unfortunately do not have access to a liquid handler to implement and describe a fully automated workflow, we add this as an additional improvement of our strategy to the discussion section and refer to selected publications to guide the readers to

further adapt their methods. In addition, we clarify in the text that the biotinylation enrichment procedure is automated in our protocol and the following steps, while performed in 96-well plate format, require manual addition of reagents.

Page 12, Lines 28-34

“While our strategy automated the enrichment of biotinylated proteins, including binding to streptavidin beads followed by rigorous washing steps, on the KingFisher Flex platform, subsequent proteolytic digestion of biotinylated proteins and clean-up for MS analysis were performed manually in 96-well plate format. Several recent studies have proposed methodologies to automate these additional steps (Fu et al, 2018; Liu et al, 2021; Müller et al, 2020), promising further reduction in hands-on time for sample preparation and increased reproducibility, albeit requiring an additional liquid handling platform.”

2. The decreased plasma membrane proteins and increased endosomal proteins during temporal serotonin trafficking as well as the transient and sustained interactions are very exciting findings that really only possible using PL-proteomics. I am curious about how these data can be used to further advance the biological understanding of serotonin trafficking and its relevant to biomedical research since serotonin can be used to treat depression and some muscle disorders. Can the authors provide some discussions on this aspect?

The reviewer raises an important point. Mapping proximal protein network changes of the activated 5HT_{2A} receptor has the potential to identify novel regulators of receptor signaling and trafficking, which are difficult to capture with other unbiased technologies, thus demonstrating a unique strength of proximity labeling proteomics. In a first step, the proximity labeling data can be leveraged to derive hypotheses for proximal interactions and their potential role in 5HT_{2A} signaling and trafficking, which can then be functional characterized and/or validated. While this would in the first place advance our biological understanding, once the interactors are functionally validated, they might provide attractive targets to finetune receptor modulation. As a motivation for choosing the 5HT_{2A} receptor as a proof-of-concept, we elude to it's importance also pharmacological target:

Page 9, Lines 6-11

“Despite the therapeutic relevance of 5HT_{2A} (Kwan et al, 2022; McClure-Begley & Roth, 2022), beyond a subset of known signal transducers and regulatory molecules, little is known about the interaction network in response to 5HT_{2A} activation. APEX-based PL allows the systematic mapping of its proximal interaction network upon receptor activation, which could unveil novel proteins regulating the receptor's activity that might in the future provide additional targets to finetune receptor activity.”

As mentioned above, we intentionally do not include more information about how activity-dependent interactions of 5HT_{2A} might affect receptor function as these would be more speculations at this point which would require functional follow up experiments. While these would indeed be interesting to perform, and we agree that it would be important to include functional studies in neuronal systems, we believe that these are beyond the scope of our manuscript, which is focused on the method and its performance.

3. From the serotonin tracking dataset, are there any key proteins that are worth following up or validate? Transient interactions are indeed difficult to validate, but authors can provide some discussions or future remarks.

We agree with the reviewer, that indeed it can be challenging to validate transient interactions. In the revised version of the manuscript, we now provide an experimental strategy to perform initial validation steps, which utilizes CRISPR-based knockout of selected interactors to perturb the interaction network followed by proximity labeling to measure the impact on the knockout on the other network components. We believe that this is very powerful as it allows dissecting which components of the interaction network might function together. For the example we knocked out two 5HT_{2A} interactors, ANXA2 and S100A10, which have been previously described to form heterotetramers that bind to membrane proteins, assisting their trafficking to the plasma membrane. Our data revealed that the knockout of both, Annexin A2 (ANXA2) and S100-A10 (S100A10) resulted in reduced biotinylation of ANXA2, S100A10, and AHNAK, which is consistent with previous findings demonstrating that these proteins physically interact. Finally, we also investigate the effect of ANXA2 and S100A10 KO on cell surface expression of the 5HT_{2A} receptor, which revealed a small but significant decrease in the expression of the 5HT_{2A} receptor at the plasma membrane in both ANXA2 and S100A10 KO cells. All of the data is provided in a new section of the manuscript including an associated main and extended data figure.

Starting Page 10, Line 16

4. Streptavidin is a major abundant contaminant in PL-proteomics data. Can the author provide the identified m/z of streptavidin peptides and compare the number of streptavidin peptides/precursors in manual vs. automated data, and in DDA vs. DIA data in supplemental figures? Since DIA fragments all masses within the wide isolation window, I am concerned that with DIA, abundant streptavidin will appear in most MS/MS scans hindering the confident identification and accurate quantification of other peptides.

We thank the reviewer for this valuable suggestion to look over the streptavidin peptides. The number of streptavidin precursors being identified in auto_DIA, manual_DIA, and auto_DDA are 12, 11, and 12, respectively, which are comparable. The detailed information regarding precursor charge, m/z, and intensity of streptavidin in the three methods were provided in **Dataset EV3**.

5. The authors carefully optimized a series of DIA parameters. Can the author also provide the rationale or optimization results that leads to the selection of 20 Da isolation window which seems large. Since PL-proteomics data is less complex compared to whole cell lysate, a smaller isolation window may benefit the accuracy of DIA identification and quantification.

We selected a 20 Da isolation window as a tradeoff between the window size, number of windows, and cycle time, to ensure that a minimum of 5 data points are collected for each peak for accurate quantification.

6. In the comparison of autoDIA vs. manualDIA, how do protein intensities correlate between autoDIA vs. manualDIA? Although we cannot compare intensities between DDA vs. DIA, it would be helpful to show how protein quantification ratios (for example APEX vs. control) correlate between DIA vs. DDA?

We looked at the correlation of protein intensities comparing autoDIA, manualDIA, and autoDDA (see figure below). The correlation comparing protein intensities for autoDIA and manualDIA is high ($R = 0.94$). Not surprisingly the intensity correlation seems to be a factor of intensity, the higher the protein intensity, the higher the correlation, while lower intensity proteins exhibit more variability. As may be expected the protein intensities from the AutoDDA method correlate to a lesser degree with either autoDIA or manualDIA ($R = 0.77$). Since DDA and DIA represent different acquisition principles, it is perhaps not surprising that the variability is higher.

Minor comments:

1. It is a valid argument to select 80 μL instead of 100 μL of beads per 1 mg of proteins to reduce nonspecific binding. However, the way to evaluate nonspecific bindings needs to be revised. More beads will of course enrich more proteins in the negative control group, also more proteins in the PM-APEX probe. The authors can provide the volcano plots or Venn diagram of PM-APEX vs control for different beads amount and highlight some nonspecific proteins in the plots. Or comparing PM-APEX 80 μL vs. 100 μL beads to show that key proteins are still enriched but maybe with more nonspecific bindings in 100 μL .

This is another helpful suggestion. We have now included a Venn diagram of PM-APEX vs. control for different volumes of Streptavidin beads in **Appendix Figure S2B**. In addition, we thought that it would be interesting to look at the intensity of Streptavidin peptides across the different bead volumes and as expected, we observe an increase in intensity with higher volumes (**Appendix Figure S2C**).

We have also edited the result section to include the discussion of these figures:

Page 6, Lines 22-34

*“To evaluate nonspecific protein binding to beads, we performed the APEX experiment for the PM-APEX cell line in the presence or absence of H_2O_2 . 60 μL of streptavidin bead slurry resulted in significantly lower peptide and protein identifications compared to 80 and 100 μL beads (**Figure***

3C). *Notably, similar numbers of proteins were identified for 80 μ l and 100 μ l bead volumes with the presence or absence of H_2O_2 (Figure 3C, Appendix Figure S2B), with the 80 μ l bead volume condition sacrificing peptide and protein IDs only minimally while achieving slightly lower nonspecific protein binding for the smaller bead volume. These results suggest that there is a wider optimum for the bead volume to be used for PL experiments. However, with a higher volume of beads, the intensities of streptavidin peptides released during proteolytic digestion also increased (Appendix Figure S2C, Dataset EV3), which might interfere with identification of biotinylated proteins. Based on these observations, we opted for 80 μ L of streptavidin beads in subsequent experiments, considering both cost-effectiveness and performance.”*

2. Can the authors label the bait proteins and maybe also several key proteins in the volcano plots in Figure 4C and provide the number of proteins used in Figure 4F for each subcellular compartment? Can the author also label the number of transient and sustained proteins in Figure 5D?

These are good suggestions.

1. We have now included enlarged volcano plots in supplemental **Appendix Figure S6** in which we include annotation for some proteins that are overrepresented for our subcellular APEX constructs compared to the control (NES-APEX cell line).
2. We have also added the number of proteins which we used for the analysis in **Figure 4F** in the figure legend: *“For the analysis we used the following number of proteins representing the different subcellular compartments: 700 for endosome/lysosome, 543 for Golgi apparatus, 1352 for plasma membrane, and 953 for cell junction”.*
3. The number of transient (n=40) and sustained (n=102) proteins were included in **Figure 5D** as well as in the figure legend: *“40 and 102 proteins are grouped into transient and sustained clusters, respectively.”*

3. Figure S2 needs to be adjusted to reveal the subtle differences of different comparison groups. For example, the authors can start the y axis with 10K or 20K peptides instead of 0. Can the author also provide the peptide IDs in Figure S2A and B?

As the reviewer suggested, in the revised version of the manuscript we have added a break of the y-axis for subpanels where we thought it would help to demonstrate the subtle differences between the comparisons. We also included the peptide IDs for the suggested figures.

Point-by-point response to Reviewer #2

In this paper, Zhong and colleagues present a high-throughput and reproducible pipeline for proximity labeling proteomics. They employ several quality-control steps during generation of clones with an APEX2-tagged protein of interest, including using flow cytometry to determine expression consistency, immunofluorescence to check tagged protein localization, and Western blot analysis of biotinylation efficiency. These cell line validation steps are relatively standard in the field and would be more rigorous if they included checking the expression level of the APEX2-tagged protein relative to the endogenous level. The authors next optimize and validate a partially automated pipeline for proximity labeling using the KingFisher Flex to perform biotinylated protein isolation and on-bead digestion and mass spectrometry sample preparation, followed by a comparison of data-independent acquisition (DIA) and data-dependent acquisition (DDA) mass spectrometry analysis. DIA in their automated format allowed more precursor ion identifications, had lower coefficients of variation, and was more reproducible, based on three replicates.

To test their pipeline's performance in subcellular proteome mapping, the authors generate separate cell lines with Lyn11 (plasma membrane), 2xFYVE (endosome), LAMTOR/LAMP1 (lysosome), GalT (Golgi), and a nuclear export signal (control) tagged with APEX2. They find many expected proteins based on GO enrichment analysis and comparison to the benchmark Human Cell Map (Go et al. 2021). Several proteins were identified in the new data not found in the Human Cell Map reference, including Golgin-45 in the Golgi, as expected based on other lines of published evidence. As a further proof-of-principle experiment this time tracking the serotonin 2A receptor-Zhong and colleagues generated HEK293T cells expressing 5HT2A-APEX2 and performed a proximity labeling time course after addition of the 5HT ligand. They find a decrease in plasma membrane protein labeling over the time course and an increase in endosomal and lysosomal protein labeling over the time course, suggesting movement of the receptor through the endocytic system. This is analogous to previous work (2017 if I remember correctly) demonstrating the use of proximity biotinylation to follow trafficking of membrane proteins (e.g. GPCRs) through the endolysosomal system, including a paper by the authors.

We thank the reviewer's comments on our study. Regarding the comment of current study being analogous to the work of Lobingier and Hüttenhain et al., 2017, we would like to clarify that the purpose of the current study is the technical aspect, demonstrating the application of the automated proximity labeling pipeline and of course we also use a different receptor in this study that we apply the APEX-based PL to. We elaborate the differences as following points:

- The purpose of the current study is to develop an unbiased proximity proteomics approach with high reproducibility and scalable throughput for the determination of subcellular proteomics and protein-protein interactions of the protein of interest (POI). By showing the mapping of proximity proteome changes of 5HT2A in HEK293T cells, we are aiming at providing an approach that is applicable to a wide range of biological applications.
- As a method, we showed the optimization of a wide range of experimental parameters for automated Kingfisher enrichment and DIA MS, leading to an optimized APEX-proximity proteomics pipeline with high reproducibility and scalable throughput. We also provided a comprehensive protocol as a step-by-step guidance for proximity proteomics starting from monoclonal cell line selection to data interpretation.

- We demonstrated the application of the pipeline to the mapping of 5HT2A receptor proximity proteome changes with stimulation of the receptor, using the APEX-tagged spatial references to demonstrate 5HT2A receptor location changes.
- In the revised version of the manuscript, we further advance the PL approach demonstrating how it can be used in combination with CRISPR-based perturbations of proximal interactors, specifically we perturbed two interactors of 5HT2A and used proximity labeling to measure the impact on the knockout on the other network components. We believe that this is very powerful as it allows dissecting which components of the interaction network might function together.

Major points:

1. The authors do not compare expression levels of APEX2-tagged proteins of interest with endogenous proteins. This is known to be an important factor in the specificity of proximity proteomics and can significantly affect the results. The authors themselves state, "both the optimal protein lysate starting amount and streptavidin bead amount might be dependent on the expression level of the PL construct and the cell type used," but they do not show a figure comparing expression levels.

Since the majority of time would be spent optimizing cell lines for expression levels, it isn't clear that the methods are as effective as advertised in speeding up analyses, although reproducibility could be enhanced with the method.

We agree with the reviewer that generating and characterizing stable cell lines expressing the APEX-fusion construct is an initial time-consuming step. We do not claim that we can speed up this part of the protocol, however once the cell lines are generated, we provide a solution to perform PL experiments with a higher throughput, with higher reproducibility, and in the revised version now also with lower sample input amounts. We believe that this is a significant improvement to the methodology and an advance which will have an impact for the community.

We also agree that achieving endogenous expression levels of the fusion construct might reduce artifacts. We do not show comparison for the APEX-fusion constructs with endogenous protein levels as our cell line does not express the 5HT2A receptor endogenously (at least to our knowledge) and the subcellular localized APEX-constructs are based on fusing APEX to a localization domain rather than an endogenous protein.

However, we add in the discussion this shortcoming:

Page 13, Lines 29-35

"Furthermore, although HEK293T cells serve as an excellent model system for demonstrating the efficacy of the automated PL approach to detect PPI dynamics with high sensitivity, these cells do not endogenously express the 5HT_{2A} receptor. Therefore, we emphasize the necessity of conducting PPI mapping in other cell types expressing endogenous 5HT_{2A}, ideally by endogenously tagging the receptor with the APEX enzyme to avoid artifacts from overexpression, and/or conducting functional characterization to validate any hypotheses derived from the proximal interaction networks."

2. The subcellular proximity proteomics experiments and serotonin receptor experiments do not appear to find any novel protein localizations, and no follow-up experiments are presented.

Furthermore, these experiments are relatively small in size, and the 96-well format using the KingFisher does not seem strictly necessary. Because the Human Cell Map was published in 2021 and used 192 baits, this paper does not appear to present a major advance.

We thank the reviewer for the comment. We would like to clarify that with our application for subcellular proteomics, we did not intend to provide a complementary study to the Human Cell Map, but rather we intend to demonstrate that our PL pipeline is applicable to these types of scientific questions. We went through our data however to point out a few location specific proteins that were discovered in our APEX study compared to Human Cell Map, but want to note that this is not the intent of this study.

We agreed that we didn't show the validation of 5HT2A receptor novel interacting proteins, which we agree is worthwhile pursuing. As a step towards it, we selected two 5HT2A interactors, S100A10 and ANXA2, to perform follow-up experiments following their knockout. Then using CRISPR knockout of these candidates, we explored (1) the network perturbation on the overall 5HT2A interaction network caused by the knockouts using our established APEX proteomics pipeline; and (2) the effects of these proteins' knockouts on 5HT2A cell surface expression, to start exploring their potential functions. The results obtained are described above and included in a completely new Results section starting on **Page 10, Line 16** and additional main and extended data figures (**Figure 6, Appendix Figure S8-9**).

3. The initial cell line generation and validation section of the paper implies that this process is somehow an advance on existing protocols or is faster in some ways than previous methods. However, the methods as described in the manuscript are relatively standard and do not seem to employ any automation that would speed up the process. Automation is not necessary, but the manuscript should be revised so as not to give the impression that this represents a new and improved cell line generation pipeline.

We agree with the reviewer that the cell line generation and validation does not represent a novel methodology and it is also not automated. When we generated multiple clonal cell lines within the course of this project, we realized that it was helpful to have a step-wise workflow for characterizing and selecting individual clones. We thought that this might be helpful information for readers as part of the method described here for scientists who are interested in implementing this method in their lab, but have no prior experience.

However, we agree that the way we presented this section in the previous version gives the impression that this cell line characterization is part of the automated workflow. To address this, we made changes in the Introduction and the Result section accordingly.

4. Numerous publications have already compared DDA- and DIA-based mass spectrometry analysis. The authors should include further discussion of how applying this to proximity proteomics presents an advance.

We added further discussion of why comparing DDA- and DIA- based mass spectrometry methods in our pipeline for APEX proximity proteomics in the Discussion section on **Page 12, Lines 20-24**.

“The consistency in quantification is of particular importance for studies across multiple conditions, for example receptor activation over a time course. In previous APEX2 studies we used a combination of DDA-based and targeted proteomics approaches to ensure consistent quantification (Lobingier et al, 2017; Polacco et al, 2024), which duplicated the measurement time compared to using the DIA approach itself.”

Minor points:

1. At several points in the manuscript, references for known protein localizations and 5HT2A interactors appear to be missing (see the last few sentences of subheading "Mapping subcellular proteomes using the automated proximity proteomics strategy" and the last paragraph of subheading "Mapping ligand-dependent proximal interaction network changes for the 5HT2A receptor").

We thank the reviewer for pointing this out. In the revised version of the manuscript we added relevant references for proteins with known cellular localizations and previously identified/studied interactors of 5HT2A.

2. The authors mention that their pipeline "could be further optimized by shortening the time for binding of the biotinylated proteins to streptavidin beads". Including data for shorter binding times and comparing it to the current overnight binding time would increase the efficiency and usefulness of this pipeline.

We agree that a shorter binding time is preferential for enhancing throughput. Therefore, we evaluated peptides and proteins being identified with binding times of 1, 2, 4, and 18 hours and provided the data in the new version of the manuscript (**Figure 2D**) and edited the result section accordingly.

Page 6, Lines 37-41

“We further aimed to enhance the throughput by testing shorter binding times for biotinylated proteins to streptavidin beads. While 1 and 2 hrs binding time seemed insufficient, minor differences were observed between 4 and 18 hrs in the number of peptides and proteins identified, indicating the feasibility of completing the enrichment of biotinylated proteins within a single day (Figure 3D).”

3. In the discussion, the authors point out that the 5HT2A receptor is not expressed in HEK293T cells used here and that another, more relevant cell type would be more informative. Perhaps

including this type of experiment would strengthen the paper if new validated interactions could be identified.

We thank the reviewer for the comment regarding the validation of 5HT2A interactions in a relevant cell type, which we agree with and therefore emphasize this in the discussion. However, given that our manuscript is a method focused paper, we decided that instead of adding additional experiments in a different cell type, we modified our PL pipeline to be amenable for low-sample input amounts, so that we can demonstrate how the PL experiments can be combined with CRISPR knockouts of 5HT2A interactors (as a proof of concept), as described in more detail above.

Validation of the interactions for example in a neuronal cell line would indeed be interesting in another study in the future.

4. APEX2-tagged protein localization shown in Figure 4A shows non-specific localization, especially of 2xFYVE. The authors should discuss why this is the case and whether localization can be improved. It is also unclear whether binding of overexpressed FYVE domains that bind to phospholipids would alter the kinetics of endosomal maturation/signaling.

In order to address this comment, we repeated the imaging analysis to look at co-localization of 2xFYVE and GalT constructs with endosomes and the Golgi apparatus, respectively. The new images are now included in the updated figure. We believe that overexposure of the initial images caused part of the non-specific localization.

To address the comment regarding potential alteration of compartmental maturation or signaling, we added a sentence in the discussion that this should be considered as a potential caveat:

Page 13, Lines 6-8

“Notably, expression of an APEX2 construct (or other proximity labeling constructs) at a subcellular compartment could potentially impact the function of the compartment.”

5. The manuscript is described as a step-by-step guide for proximity proteomics. However, several areas lack enough detail to be a true guide.

We thank the reviewer for the suggestion and agree that it would be very useful to provide a step-by-step protocol. We have addressed this comment by (1) adding additional information to the method section to clarify experimental and computational workflows and (2) putting together a detailed protocol on protocols.io: [dx.doi.org/10.17504/protocols.io.yxmvm3jbb13p/v1](https://doi.org/10.17504/protocols.io.yxmvm3jbb13p/v1)

6. The data in Figure 3 presumably only shows data at the protein level, not the precursor or peptide level. This should be stated explicitly somewhere, and it may be helpful to see precursor level data to support the author's statement that Auto_DIA outperformed the other methods in terms of precursor identification.

As suggested, we provided the data at precursor level in **Figure 3E**, peptide level in **Appendix Figure S3B and S3D**, and protein level in **Figure 3F** and **Appendix Figure S3C**.

To make the description clear, we added the following annotations to the figure legend:

Figure 3E. Performance comparison of manual enrichment protocol combined with data-independent acquisition MS (Manual_DIA), automated enrichment combined with data-

dependent acquisition MS (Auto_DDA), and automated enrichment protocol with data-independent analysis (Auto_DIA). Performance was evaluated comparing the **precursor ions** being identified from three replicates (n = 3).

Figure 3F. Distribution of the coefficient of variation of **protein intensities** from three replicates comparing Manual_DIA, Auto_DDA, and Auto_DIA approaches. Numbers depicted on the graph represent median CVs.

Appendix Figure S3B. Venn diagram analysis comparing **peptide identification** across three replicates.

Appendix Figure S3C. Venn diagram analysis comparing **protein identification** across three replicates.

Appendix Figure S3D. Correlation analysis of **peptide intensities**.

7. There seem to be only two replicates of the 2xFYVE experiment. Would the authors please explain why there are not three replicates?

We have now updated the figure legend as well as the result section stating that we only used two replicates for the 2xFYVE experiment. The reason for including only 2 replicates only is that something happened with one of them during the sample preparation, so that the data looked like an outlier.

8. The GO analysis in the right part of Figure 4E is a bit confusing since it uses the same blue color as the heat map in the left of Figure 4E. Perhaps using the color scheme in Figure 5B for Figure 4E would be more clear.

To make the GO analysis distinct to the heatmap in the left part, we changed the color scheme to green as suggested.

9. The time point used for the analysis in Figure 5C should be specified in the text and/or figure legend.

We have added the text below to the Figure 5C legend.

*"The time point of each protein used for analysis is corresponding to the maximum log₂ fold change. The intensity at each timepoint was included in **Dataset EV5**."*

10. In the discussion, the authors state the "DIA-based protein quantification already improved the reproducibility in protein identification from 78.7% to 95% overlap among three replicates...". However, it appears that reproducibility was not assessed at the protein level, so this statement is slightly misleading. Would the authors please clarify?

We agree the description was misleading. In the new version, we provided the data at precursor level in **Figure 3E**, peptide level in **Appendix Figure S3B and S3D**, and protein level in **Figure 3F and Appendix Figure S3C**.

We also revised the text as below: *"Auto_DIA outperformed Manual_DIA and DDA analysis (Auto_DDA) based on the following results: (1) The number of precursor ions identified across all the three replicates were 36,992 (95%) for Manual_DIA, 31,743 (78.7%) for Auto_DDA, and increased to 40,211 (97.1%) for Auto_DIA (**Figure 3E**). Similar trends were also observed for the peptides and proteins identified across three independent replicates (**Appendix Figure S3A-C**,*

Table EV1); (2) Auto_DIA achieved a narrower distribution of coefficients of variation (CV) for protein intensities across three independent replicates with median CV of 7%, while Manual_DIA and Auto_DDA resulted in median CVs of 13% and 11%, respectively (Figure 3F); (3) The correlation coefficients comparing peptide intensities of the three replicates ranged from 0.91-0.948 for Manual_DIA, 0.917-0.95 for Auto_DDA, but improved to 0.941-0.967 for Auto_DIA (Appendix Figure S3D)."

Comments to the editor

Overall, the pipeline presented in this manuscript presents a modest advance in proximity proteomics. This is a relatively high throughput sample processing method coupled with DIA-based mass spectrometry, but it still relies on labor-intensive cell line generation prior to proximity labeling. Furthermore, the applications presented here appear to validate the pipeline but do not uncover new information about subcellular proteomes or about serotonin signaling. In my view, a true advance would be to use the pipeline to survey hundreds of baits in order to rival the Cell Map experiment performed in the Gingras lab several years ago.

Point-by-point response to Reviewer #3

In this study, Zhong and Li et al. developed and benchmarked an automated pipeline, as well as cell line reagents, for APEX2-based spatially-resolved proteomics in cultured cells. This pipeline increased sample processing throughput and also reduced variability. The authors demonstrated its utility by profiling proteomes of several cellular compartments (inner surface of the plasma membrane, endo/lysosome, and Golgi) and temporally-resolved interactomes of the serotonin receptor 5HT2A upon agonist activation. This pipeline and dataset would be of interest to a broad community of cell biologists. Below please find specific points:

1. The authors wrote that "...while clone 1 is mainly diffused in the cytoplasm (Figure 2D), an observation also recapitulated in the wider fluorescence distribution seen by flow cytometry for this clone." A wider fluorescence intensity distribution on FACS has nothing to do with the cytosolic diffusion of APEX2 in Clone 1. It only means that the cell population is more heterogeneous with variable expression levels. The authors should not mistakenly connect them.

We thank the reviewer for the thoughtful comment and agree with it. We have removed the statement from the manuscript.

2. The authors wrote that "While 80 μ L of streptavidin beads resulted in slightly lower protein identifications compared to 100 μ L, the number of identified proteins in the negative control (without H₂O₂ treatment) was reduced using 80 μ L of beads, indicating less unspecific binding to beads (Figure S2B)." Based on Figure S2B, the difference between -H₂O₂ 80uL (2nd gray bar) and -H₂O₂ 100uL (3rd gray bar) was very small while the difference between +H₂O₂ 80uL (2nd blue bar) and +H₂O₂ 100uL (3rd blue bar) seems larger. Practically, 80uL and 100uL seem no difference overall. The authors should remove the 80/100uL comparison here.

We thank the reviewer for pointing out this question. To clarify this description, we now added a Venn diagram of PM-APEX vs. control for different volumes of Streptavidin beads in **Appendix Figure S2B**. In addition, we thought that it would be interesting to compare the intensity of streptavidin peptides across the different bead volumes and as expected, we observe an increase in intensity with higher volumes (**Appendix Figure S2C**).

We have also edited the result section to include the discussion of these figures:

Page 6, Lines 22-34

*"To evaluate nonspecific protein binding to beads, we performed the APEX experiment for the PM-APEX cell line in the presence or absence of H₂O₂. 60 μ l of streptavidin bead slurry resulted in significantly lower peptide and protein identifications compared to 80 and 100 μ l beads (**Figure 3C**). Notably, similar numbers of proteins were identified for 80 μ l and 100 μ l bead volumes with the presence or absence of H₂O₂ (**Figure 3C, Appendix Figure S2B**), with the 80 μ l bead volume condition sacrificing peptide and protein IDs only minimally while achieving slightly lower nonspecific protein binding for the smaller bead volume. These results suggest that there is a wider optimum for the bead volume to be used for PL experiments. However, with a higher volume of beads, the intensities of streptavidin peptides released during proteolytic digestion also increased (**Appendix Figure S2C, Dataset EV3**), which might interfere with identification of*

biotinylated proteins. Based on these observations, we opted for 80 μ L of streptavidin beads in subsequent experiments, considering both cost-effectiveness and performance.”

3. In Figure 4A, the 2xFYVE GFP staining has a much broader expression compared with Rab5, indicating non-endosome localization. The authors should comment on this and the potential caveats caused by it. The GalT (Golgi) one has a similar problem but is likely less severe.

In order to address this comment, we repeated the imaging analysis to look at co-localization of 2xFYVE and GalT constructs with endosomes and the Golgi apparatus, respectively. The new images are now included in the updated figure. We believe that overexposure of the initial images caused part of the non-specific localization. In order to remove potential nonspecific labeling identifications from our proximity labeling data, we perform a ratiometric analysis in which we compare the labeled proteins of each compartment-specific APEX2 construct to the proteins labeled by the cytosolic NES-APEX2 construct and only describe the proteins as compartment specific, which are overrepresented compared to the NES-APEX2.

4. As a resource/method paper, the authors should consider depositing the constructs and cell lines made in this study to Addgene or other repositories.

We deposit the constructs on Addgene and the Addgene IDs of plasmids are included in **Dataset EV1**.

Point-by-point response to Reviewer #4

Review notes for: An automated proximity proteomics pipeline for subcellular proteome and protein interaction mapping in Molecular Systems Biology

Summary:

This paper presents a workflow for the preparation and analysis of APEX2-based proximity interaction proteomics. The aim is to construct a framework for other researchers to quickly perform proximity-dependent experiments. The core of the workflow is a method using a KingFisher liquid handler to do biotinylated protein enrichment. This is followed by DIA-MS analysis as it was shown to be superior to traditional DDA analysis. The workflow is demonstrated by confirming previous Cell Map localizations and then by constructing a time-course of the response of a serotonin receptor after activation by 5-hydroxytryptamine.

General Remarks:

The study's outcomes align with prior research and make sense, with the authors providing plausible interpretations. The paper showcases a technical progression and could serve as a valuable resource for researchers aiming to enhance the throughput of their proximity-labeling datasets. The biological findings, however, are not as prominent as the technical accomplishments.

In conclusion, while this manuscript is acceptable in principle (once the issues below are addressed), it is in a nascent stage. It leaves much to be desired due to the unexplored potential within the premise of an "automated method". Indeed, the workflow's automation is minimal, predominantly focusing on a single portion of the experiment. Several upstream molecular biology steps, crucial for obtaining good results, are left manual and could greatly benefit from automation. The title, promising an automated proximity biotinylation workflow, is somewhat misleading as it implies more comprehensive automation than what is presented in the study.

Major points:

1. Numerous times throughout the manuscript, including the title, refer to the presented pipeline as 'The manuscript repeatedly designates the proposed pipeline as 'automated' in several instances, including the title. However, this is somewhat misleading as only a fraction of the pipeline is automated. The bulk of the procedure, including steps like cell lysis (and possibly digestion-it's unclear), which could feasibly be conducted on the liquid handler, remains manual. Consideration of the cell line generation in an automated or semi-automated manner is not presented.

We agree with the reviewer that the previous version of the manuscript might have suggested additional automated experimental steps. We have carefully revised the manuscript to address the reviewer's concern:

1. The manuscript is now entitled "*A proximity proteomics pipeline with improved reproducibility and throughput*", which we feel does more accurately describe the essence of the manuscript.
2. While we do not have a liquid handling device available in our lab, we have added information and relevant references suggesting which additional steps of the protocol could be automated based on previously described work, including the digestion and clean up of the samples.

3. In addition, we clarify in the text that the biotinylation enrichment procedure is automated in our protocol and the following steps, while performed in 96-well plate format, require manual addition of reagents.
4. In several parts of the manuscript which previously stated the automated proximity labeling approach, we adjusted the text to reflect which experimental part is automated.

Notably, while our method does not automate every possible step of the proximity labeling protocol from cell lysis to the analysis by the mass spectrometer, we truly believe based on our own experience in applying this protocol, that we provide a significant advance for performing proximity labeling experiments at scale. While manual enrichment allows at most processing 20 samples in parallel with a lot of hands on time for the individual washing steps, using our protocol it is now possible to process 96 samples simultaneously with very little hands on time (**Figure 3G**).

2. Furthermore, the power of an automated platform for method optimization is almost completely neglected. For example, only a single protein load was tested against a handful of streptavidin particle amounts. There was a desire to reduce the background intensity of non-specific captured proteins, yet only a single very long initial binding time (12 hours) was explored. This is disappointing. An automated platform allows a broad range of different parameters to be tested with ease, but in this case, only very limited ones are attempted. It might be possible that the authors conducted a meticulous optimization of parameter selection before their work, but this isn't presented in the manuscript. If this is the case, I suggest including this data to assist other researchers in identifying key parameters.

We thank the reviewer for this suggestion. To demonstrate the power of the automated platform we have expanded our manuscript with two additional completely novel datasets:

1. In the revised version of the manuscript we have now extended our method optimization and explored a range of binding times (1, 2, 4, and 18 hours) (**Figure 3D**). The data shows that 4 hours of binding is sufficient to achieve comparable results to 18 hours binding. The shorter binding time allows performing the binding and washing step within one day thus shortening the overall protocol from cell lysis to samples ready for proteomic analysis to two days.
2. To further demonstrate the power of the automated platform, we have (1) downscaled the protocol to be amenable for lower sample input so that the cell culture experiments can now be performed in 6-well plate format (**Figure 6A-C**). We believe that the smaller scale input material is the perfect combination with the automated enrichment of biotinylated proteins in 96-well plate format and in fact we demonstrate that the performance is minimally affected despite the low input material. To showcase this advance across multiple conditions, we selected two proteins, S100A10 and ANXA2, of the activity-dependent 5HT2A proximal interaction network, performed CRISPR knockouts of these factors, and monitored how these knockouts affected other network components (**Figure 6D-G**). For each of the genes, we included two separate gRNA for the knockout, 2 non-targeting controls, 3 different timepoints following receptor activation, as well as 3 biological replicates of each condition, which are in total 54 samples which were processed in parallel.

3. The choice and presentation of the mass spectrometry methods are less than ideal. For instance, during their method optimization, they analyze DIA data in a library-free fashion. However, for their actual biological experiments, they depend on spectral library-based analysis. The reasoning behind this decision is unclear, as the results from a library-free analysis can significantly differ from those where a spectral library limits the types of peptides discovered.

We thank the reviewer for pointing this out. We agree that there can be differences in protein identification when using a library-free and library-based workflow. We observed that at least in our hands a slightly higher consistency in precursor identifications across runs when we apply a library-based analysis (compare figures below: top graph summarizes the library-based results compared to the library-free results bottom). Therefore, we felt that it is worthwhile for the biological samples to analyze a subset of the samples in DDA mode to generate a spectral library. Since the generation of a spectral library requires additional mass spec runs, we decided that these are less critical for the optimization of the protocol as we are not aiming to discover new biology from these samples, but rather perform a relative comparison between different conditions. In order to clarify this for the reader of our manuscript, we have added the following descriptions in the **Method section**.

Page 24, Lines 4-9

“In the data analysis of methods optimization, the DIA datasets were searched against the SwissProt Human database (downloaded 10/2020) by using a spectral library-free approach (direct-DIA) in Spectronaut (version 16.0). We observed a slightly higher consistency in precursor identifications across runs when comparing library-free and library-based analysis of subcellular proteomes, so in the data analysis of mapping subcellular proteomes or protein interaction dynamics, the DDA datasets were collected to construct a spectral library.”

4. The absence of statistical comparisons in their initial method optimization results is puzzling, especially given the effective application of statistics to their proximity labelling data. They have three methods with three replicates but only report the mean (presumably-it's not specified) for the various numbers of identifications. Why not include the coefficient of variation (CV), and potentially compare these three methods statistically using a basic tool like the Students' t-test?

We thank the reviewer for this suggestion. We have addressed this comment in multiple ways:

1. We included statistical comparisons for the method optimizations using Students' t-test (see **Figure 3C-D, and Appendix Figure S3A**).

2. We provided the information regarding the number of replicates for each subpanel and the statistical test performed.
3. We provided the coefficient of variation of **protein intensities** from three replicates comparing these three methods in **Figure 3F**.
4. We provided **Table EV1** to include details of precursor features identification for each replicate in each of the three methods.

Minor point:

The language surrounding the MS acquisition and analysis parameters could be made clearer. For example, 1) the samples were dried by vacuum centrifugation, but they must have been resuspended in something to be injected on the LC. What was used for this resuspension? How much of the sample was injected? Was the injection performed directly from the well plate? 2) Was the digestion process carried out using the liquid handler? I suggest that the authors carefully revisit the presentation of their methods.

We thank the reviewer for pointing this out, which we agree is important for a method paper. In the revised version of our manuscript we expanded the Method section to include missing experimental details. Specifically, 1) we added “Samples were dried by vacuum centrifugation, resuspended in 20 μ L 0.1% formic acid, and injected 1 μ L for MS analysis.” in the Method section, and provided **Table EV2** to include analysis parameters. 2) we do not use liquid handler, so we added explanation “While our strategy automated the enrichment of biotinylated proteins, including binding to streptavidin beads followed by rigorous washing steps, on the KingFisher Flex platform, subsequent proteolytic digestion of biotinylated proteins and clean-up for MS analysis were performed manually in 96-well plate format.” in the Discussion section. Moreover, we put together a step-by-step online protocol ([dx.doi.org/10.17504/protocols.io.yxmvm3jbb13p/v1](https://doi.org/10.17504/protocols.io.yxmvm3jbb13p/v1)), which we also refer to in the Method section of the manuscript.

5th Jun 2024

Manuscript Number: MSB-2023-11697R

Title: A proximity proteomics pipeline with improved reproducibility and throughput

Dear Ruth,

Thank you for sending us your revised manuscript. We have now heard back from two of the three reviewers who agreed to evaluate your revised study. Reviewer #2 was not available this time around. In the interest of time, we decided to proceed with making a decision based on the two available reports. If we hear from reviewer #4 in the next few days, we will forward their comments. As you will see below, the two reviewers think that the performed revisions have addressed their major concerns. However, reviewer #1 still raises a few concerns, which we would ask you to address in a revision. All issues can be addressed by text modifications, no additional analyses are required. Regarding the title change requested by reviewer #1, we think that it is OK to leave the title as it is (i.e. without referring specifically to the application to the serotonin receptor).

We would also ask you to address some minor editorial issues listed below.

- Our data editors have noticed some unclear or missing information in the figure legends, please address the following:
 - Please define the annotated p values ****/** in the legend of figure 3d; as appropriate.
 - Please indicate the statistical test used for data analysis in the legend of figure 5a.
 - Please include the information related to n in the legends of figures 3f; 6d.
 - Please describe the nature of entity for 'n' (biological? technical?) in the legends of figures 3c-d.
 - Please define the error bars in the legends of figures 3c-d; 6d.
- The Figure Legends should be included after the References.
- Please include the corresponding author's email address on the title page of the manuscript text.
- The two Tables included in the Appendix file (currently called Table EV1 and Table EV2) need to be renamed Appendix Table S1 and Appendix Table S2. Please make sure that their callouts in the text are updated accordingly.
- The funding information provided in the manuscript text (Acknowledgements) should match the information entered in the online submission system. Currently the NIH grant R37DA045657 and the grants from the UCSF Program for Breakthrough Biomedical Research funded in part by the Sandler Foundation and the UCSF Research Resource Fund Award are missing from the submission system.
- Please include 5 keywords.
- Please remove the 'Authors Contributions' from the manuscript. The 'Author Contributions' section is replaced by the CRediT contributor roles taxonomy to specify the contributions of each author in the journal submission system. Please use the free text box in the 'author information' section of the online submission system to provide more detailed descriptions if needed (e.g., 'X provided intracellular Ca⁺⁺ measurements in fig Y').

Please resubmit your revised manuscript online, with a covering letter listing amendments and responses to each point raised by the referees. Please resubmit the paper ****within one month**** and ideally as soon as possible. If we do not receive the revised manuscript within this time period, the file might be closed and any subsequent resubmission would be treated as a new manuscript. Please use the Manuscript Number (above) in all correspondence.

Click on the link below to submit your revised paper.

Kind regards,

Maria

Maria Polychronidou, PhD
Senior Editor
Molecular Systems Biology

If you do choose to resubmit, please click on the link below to submit the revision online before 5th Jul 2024.

IMPORTANT:

Please note that corresponding authors are required to supply an ORCID ID for their name upon submission of a revised manuscript (EMBO Press signed a joint statement to encourage ORCID adoption).

(<https://www.embopress.org/page/journal/17444292/authorguide#editorialprocess>)

Currently, our records indicate that the ORCID for your account is 0000-0002-0896-5910.

Link Not Available

*** PLEASE NOTE *** As part of the EMBO Press transparent editorial process initiative (see our Editorial at <https://dx.doi.org/10.1038/msb.2010.72> , Molecular Systems Biology will publish online a Review Process File to accompany accepted manuscripts. When preparing your letter of response, please be aware that in the event of acceptance, your cover letter/point-by-point document will be included as part of this File, which will be available to the scientific community. More information about this initiative is available in our Instructions to Authors. If you have any questions about this initiative, please contact the editorial office (msb@embo.org).

Reviewer #1:

The authors have addressed most of my comments. A new section and more figures were added to demonstrate the applicability of the PL pipeline to lower input samples using 6-well cell culture scale instead of 10 cm dish scale. And the authors further knocking out proteins in the serotonin interaction network, including ANXA2 and S100A10, to understand perturbations to the serotonin network.

Automating proximity labeling proteomics workflow is important because the traditional workflow is very tedious, expand through multiple days, and with limited throughput when using the traditional magnetic rack. So, to enable reproducible and large-scale biological comparisons, automating PL workflow, even if partly, is important for the field. I therefore disagree with reviewer 2 about the necessity to automate this workflow. Automating proteomics workflow is not a trivial task and the authors have conducted many optimizations to achieve this goal, at least for the enrichment and washing steps, which is the most labor-intensive steps. The author also provided a step-by-step protocol on protocols.io, which will be helpful to the field.

Although limitations still exist in this study, the authors have provided objective discussions to these limitations. The manuscript could be acceptable after addressing some of my additional comments below:

1. The authors have demonstrated the applicability of this PL pipeline for 10 cm and 6-well dish scale of cell culture material. Can the author comment on the possibility of further reducing the sample amount, for example to 12-well dish or 24-well dishes? Other studies claiming low-input samples are typically at the 96-well plate scale. So be careful of claiming 6-well dish as low-input. Proximity labeling does require more materials since there is an enrichment step. But it would be helpful to know what factors should be considered when attempting to lower down the starting material.
2. Can the author provide the rationale to select ANXA2 and S100A10 from other identified protein networks? Why not selecting the GPCR signaling proteins in Figure 5E?
3. Since ANXA2 and S100A10 KO are important proteins in the serotonin network, it is surprising to find that neither ANXA2 nor S100A10 KO resulted in significant changes in sustained, activity-dependent proximal interactions. Can author provide more discussions about this?
4. The authors have recognized that more future follow up studies should be conducted to validate and follow up on the serotonin receptor biology. But it is unclear how and what future studies can be conducted and what would be the goal and impact? Can the authors provide more discussions in this regard?
5. The authors have recognized the overstatement of fully automated workflow in the title and manuscript. However, the new title

"A proximity proteomics pipeline with improved reproducibility and throughput" seems to be very vague. It does not address any serotonin-related results or the temporal changes of receptor activation. A thorough cell line generating results section was included in this study but was not reflected in the title. I think the PL experiments and the temporal data analysis strategy in this study would be helpful to guide the design of other receptor activation experiments in the field. So these aspects could be reflected in the title.

Reviewer #3:

The authors have addressed my concerns.

All editorial and formatting issues were resolved by the authors.

11th Jun 2024

Manuscript number: MSB-2023-11697RR

Title: A proximity proteomics pipeline with improved reproducibility and throughput

Dear Ruth,

Thank you again for sending us your revised manuscript. We are now satisfied with the modifications made and I am pleased to inform you that your paper has been accepted for publication.

Kind regards,
Jingyi

Jingyi Hou, PhD
Scientific Editor
Molecular Systems Biology
